# Single-cell spatial metabolomics with cell-type specific protein profiling for tissue systems biology

Thomas Hu[1,2,10], Mayar Allam [1,10], Shuangyi Cai[1], Walter Henderson [3], Brian Yueh[4], Aybuke Garipcan[4], Anton V. Ievlev [5], Maryam Afkarian [6], Semir Beyaz[4] & Ahmet F. Coskun [1,7,8,9] ✉

Metabolic reprogramming in cancer and immune cells occurs to support their increasing energy needs in biological tissues. Here we propose Single Cell Spatially resolved Metabolic (scSpaMet) framework for joint protein-metabolite profiling of single immune and cancer cells in male human tissues by incorporating untargeted spatial metabolomics and targeted multiplexed protein imaging in a single pipeline. We utilized the scSpaMet to profile cell types and spatial metabolomic maps of 19507, 31156, and 8215 single cells in human lung cancer, tonsil, and endometrium tissues, respectively. The scSpaMet analysis revealed cell type-dependent metabolite profiles and local metabolite competition of neighboring single cells in human tissues. Deep learning-based joint embedding revealed unique metabolite states within cell types. Trajectory inference showed metabolic patterns along cell differentiation paths. Here we show scSpaMet's ability to quantify and visualize the cell-type specific and spatially resolved metabolic-protein mapping as an emerging tool for systems-level understanding of tissue biology.

Spatially resolved metabolomic analysis of human tissues is paramount for the study of chemical balances and alterations in health and disease. Metabolites and lipids play a regulatory role in immune responses and cancer[1,2]. Particularly, metabolism in tumors has demonstrated vital mechanisms in understanding the functional changes in immune and cancer cell interactions[3]. Immune cell types experience significant metabolic programming when infiltrating the tumor ecosystem[4–6]. Cancer progression controls multiple immune and stromal cell types and their metabolic functional roles in tumors due to the rapid nutrient depletion and accumulation of waste products during rapid proliferation[7].

Thus, it is crucial to identify cell types and their metabolism in biological tissues.

Recent advances in Mass spectrometry imaging (MSI) techniques have allowed spatial profiling of a large number of proteins and metabolites. MSI methods have provided capabilities to capture metabolite information within its spatial context to address the loss of spatial details in bulk-level mass spectrometry techniques[8]. Several technological advancements in mass spectrometry imaging have allowed the acquisition of spatial metabolite features including matrix-assisted laser desorption/ionization (MALDI), desorption electrospray ionization (DESI), and secondary ion mass spectrometry (SIMS)[9]. MSI

[1]Wallace H. Coulter Department of Biomedical Engineering, Georgia Institute of Technology and Emory University, Atlanta, GA, USA. [2]School of Electrical and Computer Engineering, Georgia Institute of Technology, Atlanta, GA, USA. [3]Institute for Electronics and Nanotechnology, Georgia Institute of Technology, Atlanta, GA, USA. [4]Cold Spring Harbor Laboratory, Cold Spring Harbor, NY, USA. [5]Oak Ridge National Laboratory, Center for Nanophase Materials Sciences, Oak Ridge, TN, USA. [6]Division of Nephrology, Department of Internal Medicine, University of California, Davis, CA, USA. [7]Interdisciplinary Bioengineering Graduate Program, Georgia Institute of Technology, Atlanta, GA, USA. [8]Winship Cancer Institute, Emory University, Atlanta, GA, USA. [9]Parker H. Petit Institute for Bioengineering and Bioscience, Georgia Institute of Technology, Atlanta, GA, USA. [10]These authors contributed equally: Thomas Hu, Mayar Allam. ✉e-mail: ahmet.coskun@bme.gatech.edu

methods characterized metabolic heterogeneity in tissue samples with different sensitivity, spatial resolution, and chemical coverage. MALDI and DESI efficiently map metabolites at 10-100 μm spatial resolution while Time-of-Flight Secondary Ion Mass Spectrometry (TOF-SIMS) acquires lipids and metabolic fragments at sub-micron spatial details[10–12]. However, these MSI approaches lacked cell type tagging, causing the loss of key cell-specific correlates in tissues.

Characterization of the single-cell level metabolomic profile remains a difficult task. A recently proposed method called SpaceM[13] performed an integrated analysis of MALDI-based metabolic maps and fluorescence microscopy images in single cultured cells but lacked high-resolution details of "true" single cells in dense tissues. Instead, SpaceM mapped out pixelated correlations of metabolic targets within cytosolic boundaries of large cells in cultures. Another method called SEAM[14] was proposed for single nuclear metabolomic profiling in the native tissue microenvironment using TOF-SIMS. While this method has allowed submicron metabolite mapping in cells and tissues, the association of specific cell types to the metabolic profiles has been lacking and only nuclei patterns are extracted from the entire cells without the cytosolic boundaries. On the other hand, Imaging Mass Cytometry (IMC) has provided multiplex imaging of 35 protein markers in the spatial context of patients' samples at subcellular resolution (1-μm)[15–17], albeit without metabolic target information. Recently, a three-dimensional (3D) spatially resolved metabolic profiling framework (3D-SMF)[18] was demonstrated to incorporate isotope tagging of cell-type specific channels measured by TOF-SIMS. While this method showed promising results for detecting correlations of cell types with metabolic channels, the capability of achieving single-cell details of joint protein and metabolite analysis has not fully been realized in spatially crowded human tissues.

Several algorithms have been developed for the integration of multi-omics data such as MNN[19], Scanorama[20], Conos[21], MARIO[22], Seurat[23], LIGER[24], and SEAM[14]. MNN, Scarnorama, and Conos provide computational pipelines for batch effect removal from multiple datasets. MARIO integrates multi-omics datasets by matching with partial overlap accounting for both shared and distinct features. Seurat uses an unsupervised framework to integrate multi-omics datasets by assigning relative weights of each data type in each cell. LIGER formulates an integrative nonnegative matrix factorization problem to address multi-omics data integration from different modalities and protocols. While multi-omics integration methods are being developed, spatial metabolomic data are unsuitable for integration with other modalities because of their untargeted nature, lack of shared markers, and identified cell types. On the other hand, single-cell analysis methods such as VEGA[25], scDHA[26], scMM[27], SPADE[28], DPT[29], Squidpy[30], Athena[31], and SPEX[32] were developed for latent embedding generation, cell lineage reconstruction, and spatial analysis. Using an autoencoder model, VEGA, scDHA, and scMM proposed methods for extracting single-cell level latent variables suitable for downstream analysis such as clustering and visualization. SPADE and DPT are methods developed for inferring single-cell developmental progress from data. Squidpy, Athena, and SPEX are a suite of algorithms for analyzing spatial omics data by introducing graph construction from single-cell spatial data and characterization of cell type neighboring frequency and spatial properties. While those methods showed great progress in the analysis of proteomics data, they are not tailored for single-cell metabolomics and no work is focusing on single-cell competition[33,34]. It is therefore important to introduce specific analysis pipelines adapted for joint metabolomic and proteomic datasets.

To provide a complementary solution to the need for simultaneous whole-cell metabolic and protein analysis in situ, we propose Single Cell SPAtially resolved METabolic (scSpaMet) framework for profiling immune cells and cancer cells in human tissues at the single-cell level by incorporating untargeted spatial metabolomics and targeted multiplexed protein imaging in a single pipeline. scSpatMet combines previously developed 3D-SMF[18], a framework that achieves submicron resolution for metabolic imaging, with multiplex IMC proteomic imaging for cell-type characterization in the same tissue sample. scSpaMet enables the correlation of more than 200 metabolic markers and 25 protein markers in individual cells within native tissues. Moreover, scSpaMet introduces additional analysis capabilities for joint metabolomic and proteomic single-cell data.

## Results

With the advent of the current immunotherapy approaches, it is becoming critical to develop a comprehensive understanding of immune metabolism. The multi-omics scSpaMet approach has the great potential to link the multi-layer information of the proteomics data with the metabolism data on the same biological tissue. The scSpaMet starts with staining the tissues with the metal-isotope conjugated antibodies, performing the metabolic profiling using the ToF-SIMS imaging, and finally performing the proteomic profiling using IMC. 3D-SMF[18] was developed to profile hundreds of metabolic fragments' mass spectrum peaks in tonsils using ToF-SIMS at the tissue level and the protein expression profile at the single cell level of immune cells in tonsil and lung tissues using IMC[35,36]. Every multiplexed imaging region in the SIMS data has a resolution of better than 1 μm per pixel for over 200 m/z peaks. Further, IMC provides targeted multiplex protein imaging data for deciphering distinct cell types (for instance, cancer/epithelial, stroma, and immune cells) at 1 μm per pixel resolution for up to 40 markers. Compared to existing metabolomic profiling methods, scSpaMet allows correlation of multiplex cell types to metabolic profile at the single-cell level. Compared to 3D-SMF, the scSpaMet imaging pipeline incorporated in situ sequential detection of metabolomic and proteomic within the same tissue (Supplementary Fig. 1a and Supplementary Table 1), providing correlative proteomics/metabolomics analysis at the single-cell level by cross-modality spatial registration (Supplementary Fig. 1b). Accurate single-cell segmentation from the scSpaMet pipeline allowed single-cell level joint metabolite and protein downstream analysis, whereas 3D-SMF only allowed metabolite channel-level correlation, channel embedding, and pixel clustering from tissue regions (Supplementary Fig. 1c).

In the scSpaMet pipeline, the sequential ToF-SIMS and IMC datasets were combined and matched to the single cell level to integrate the information and perform comparative analysis (Fig. 1a). The scSpaMet was used to dissect the metabolism in lung tumors (Fig. 1b) and tonsil tissues (Fig. 1c). First, a consecutive tissue slide is stained separately using Hematoxylin and Eosin (H&E) to identify the imaging region of interest before scSpaMet profiling and downstream analysis (Supplementary Figs. 2a and 3 and 4). Next, sequential ToF-SIMS and IMC imaging procedures are performed to extract spatial maps of metabolites and proteins. Pixel clustering of SIMS data reveals unique metabolic variation in the spatial context (Supplementary Figs. 5–7 and Methods). To quantify cell-type specific metabolic profiles, a cross-modality single-cell registration pipeline was developed utilizing Histone 3 and Intercalator markers in the IMC dataset, and Phosphate 79 m/z channels in TOF-SIMS dataset, allowing the joint analysis of protein-metabolite modalities in single cells (Fig. 2, Supplementary Figs. 8–10 and Methods). Using affine transformation, the cross-modality pipeline yields higher structural similarity (SSIM) and normalized root mean square error (NRMSE) compared to rotation only and random shift. The registration quality was quantified using the same metrics of SSIM and NRMSE. Single-cell segmentation was used to extract the protein and metabolite expression levels and their spatial locations.

Due to the untargeted discovery nature and low variability nature of metabolomic data (Fig. 3a), it is important to develop and identify suitable analysis tools. Existing computational pipelines for integrative analysis of single-cell data are developed for (1) same-modality integration[19–21], (2) cross-modality integration with shared features[22,23], (3) cross-modality multi-modal on same cell[23,24] (Fig. 3b).

The first data integration (1) is for the same type of sequencing modality for batch effect removal in the data. This is also applied to each modality in our metabolomic and proteomic data. The second type of algorithm (2) works under the assumption that multi-omics data share the same marker set to some extent and multi-omics integration is achieved by finding matching cells in the overlapping feature space across data. The third type of analysis work (3) is for sequencing on the same cell under the assumption of matching cell types across modalities in multimodal analysis. These methods achieve successful integration by looking at shared markers or cell types. Our proteomic/metabolomic single-cell data does not share the same modality nor marker set. Moreover, single-cell metabolomic profiles are less variant

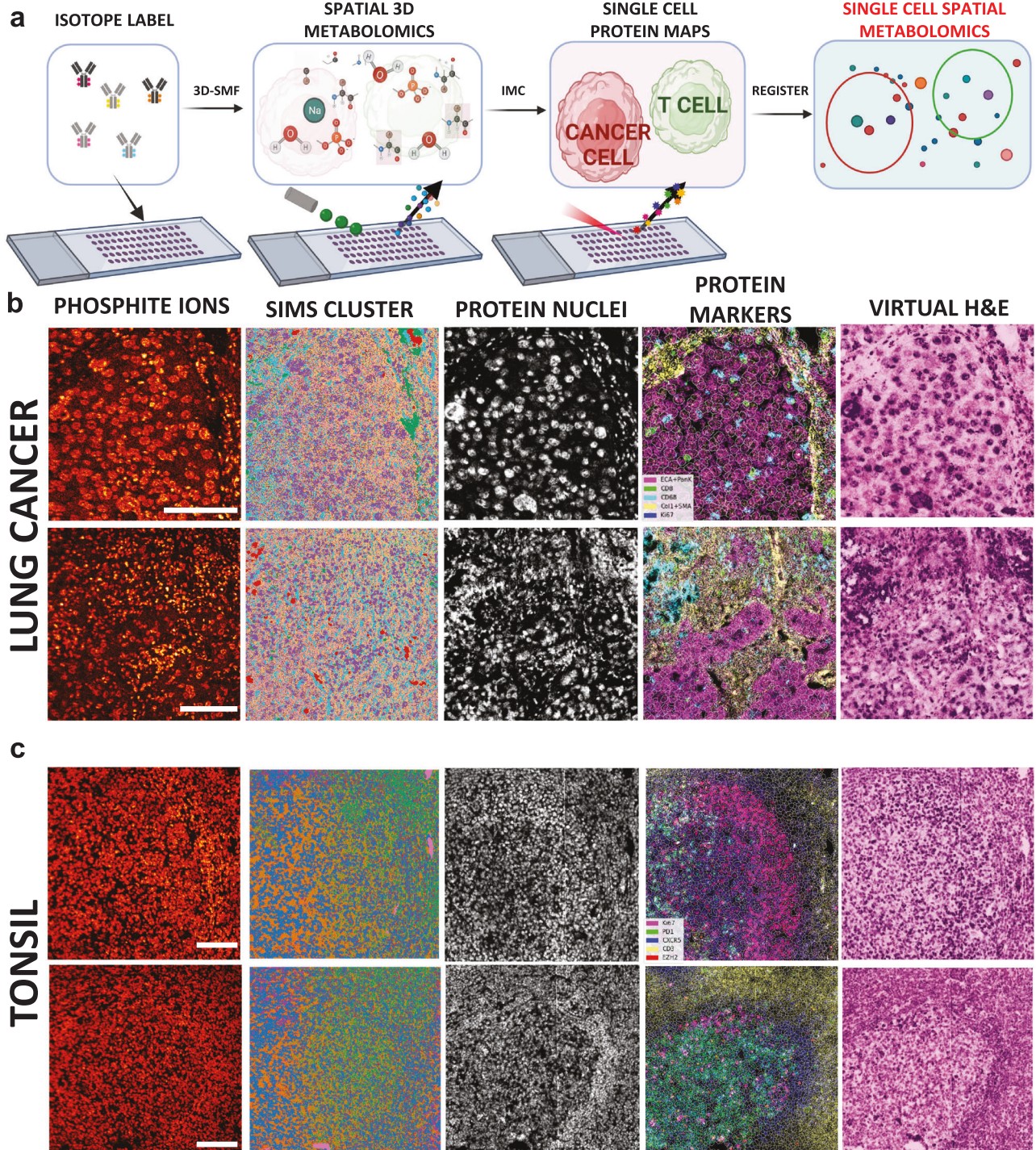

**Fig. 1 | The scSpaMet pipeline for integrated metabolite and protein profiling at the single-cell resolution. a** Overview of scSpaMet. Tissue samples on glass slides are labeled with metal-isotope conjugated antibodies followed by metabolic profiling with 3D-SMF and finally proteomic profiling using IMC. Created with Bioender.com. **b** Examples of scSpaMet generated data in lung cancer tissues. Left to right: PO3- channel ion, multiplexed metabolic data processed by pixel clustering, IMC imaging Histone H3 marker, multiplexed IMC data overlaid with pseudo-coloring, virtual reconstructed H&E staining from IMC multiplexed proteomic data. *n* = 7 biologically independent samples on 21 FOVs. Scale bar 100 μm. **c** Examples of scSpaMet generated data in tonsil tissues. Left to right: PO3- channel ion, multiplexed metabolic data analyzed by pixel clustering, IMC imaging of Intercalator marker for DNA, multiplexed IMC proteomic data overlaid with pseudo-coloring, virtual reconstructed H&E staining from IMC multiplexed data. *n* = 2 biologically independent samples on 11 FOVs. Scale bar 100 μm.

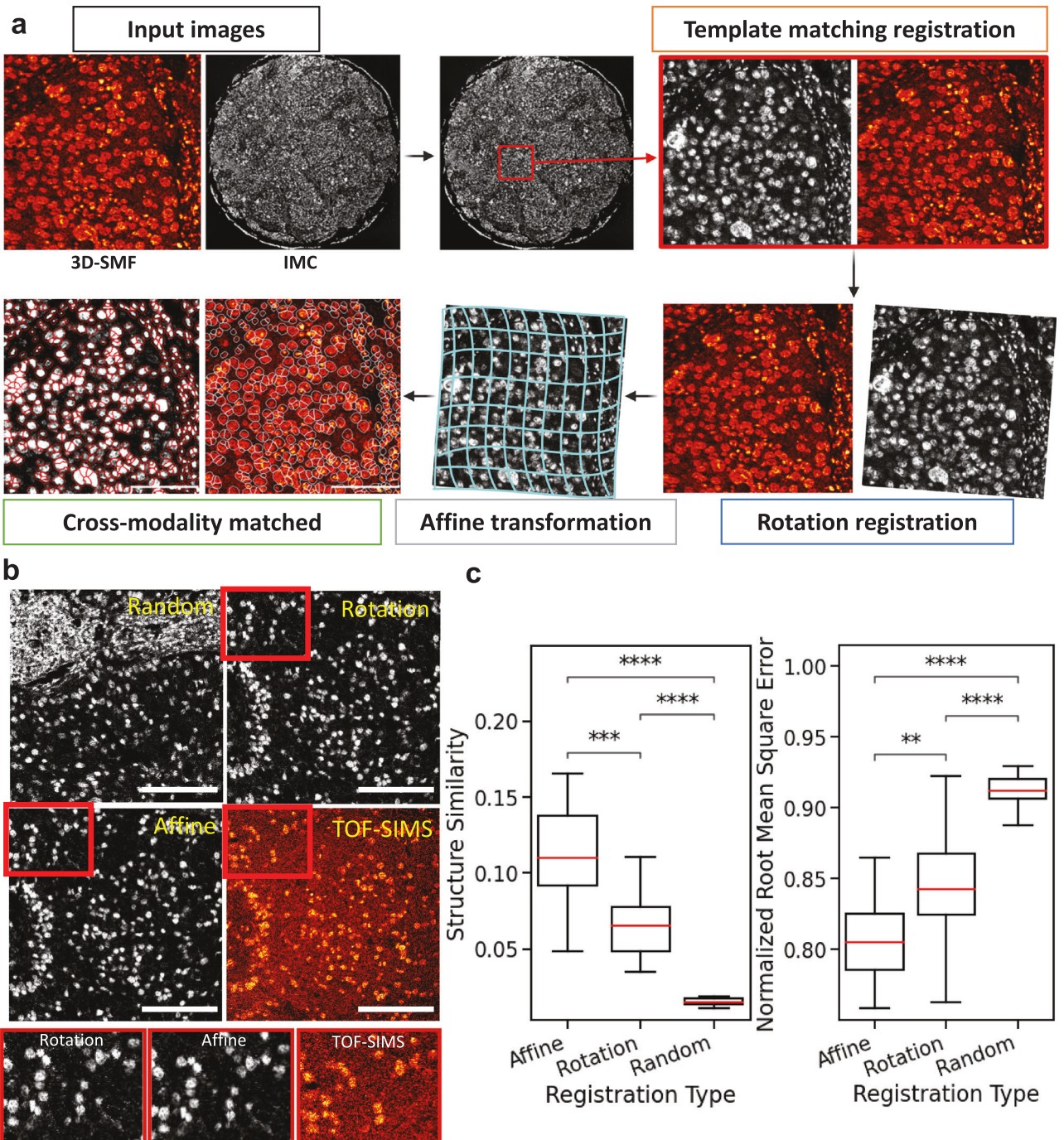

**Fig. 2 | Single-cell cross-modality registration pipeline. a** Single-cell protein-metabolite bi-modal registration pipeline. Input images consist of 3D-SMF and IMC-generated images. First, a template matching algorithm is used to find the corresponding matching region between the 3D-SMF region (smaller) inside the IMC region (larger). Next, the rotation offset between the two aligned and cropped images is calculated. Finally, the affine transformation of the two images is calculated to obtain bi-modal matched images. **b** Registration result from the comparison between random, rotation registration, and affine registration. Left: Examples of the three registration methods between IMC (gray) and 3D-SMF (red) images and their corresponding inset. Scale bar 100 μm. **c** Comparison of Structural/structure Similarity (SSIM) and Normalized Root Mean Square Error (NRMSE) between the three registration methods ($n = 24$ FOVs). Mann–Whitney-Wilcoxon test two-sided with Bonferroni correction (ns: $0.05\ p$, *: $0.01\ p <= 0.05$, **: $0.001\ p <= 0.01$, ***: $0.0001 < p <= 0.001$, ****: $p <= 0.0001$). All box plots with center lines showing the medians, boxes indicating the interquartile range, and whiskers indicating a maximum of 1.5 times the interquartile range beyond the box.

across cell types, therefore unsuitable for using algorithms on integration for same cell sharing cell types. (Supplementary Table 2).

The scSpaMet pipeline provides three additional analysis capabilities for the understanding of multimodal single-cell level data combining protein markers and metabolite channels: (1) muti-omics cell competition (Supplementary Fig. 2b), (2) multi-modal data integration (Supplementary Fig. 2c) and (3) multi-omics trajectory inference (Supplementary Fig. 2d). In the multi-omics cell competition pipeline (1), we extracted cell type, cell metabolomics profile, and their spatial location and compared local neighboring cell metabolomic profile variation to infer the local competition of metabolite at the single cell level. In the multi-modal data integration pipeline (2), we

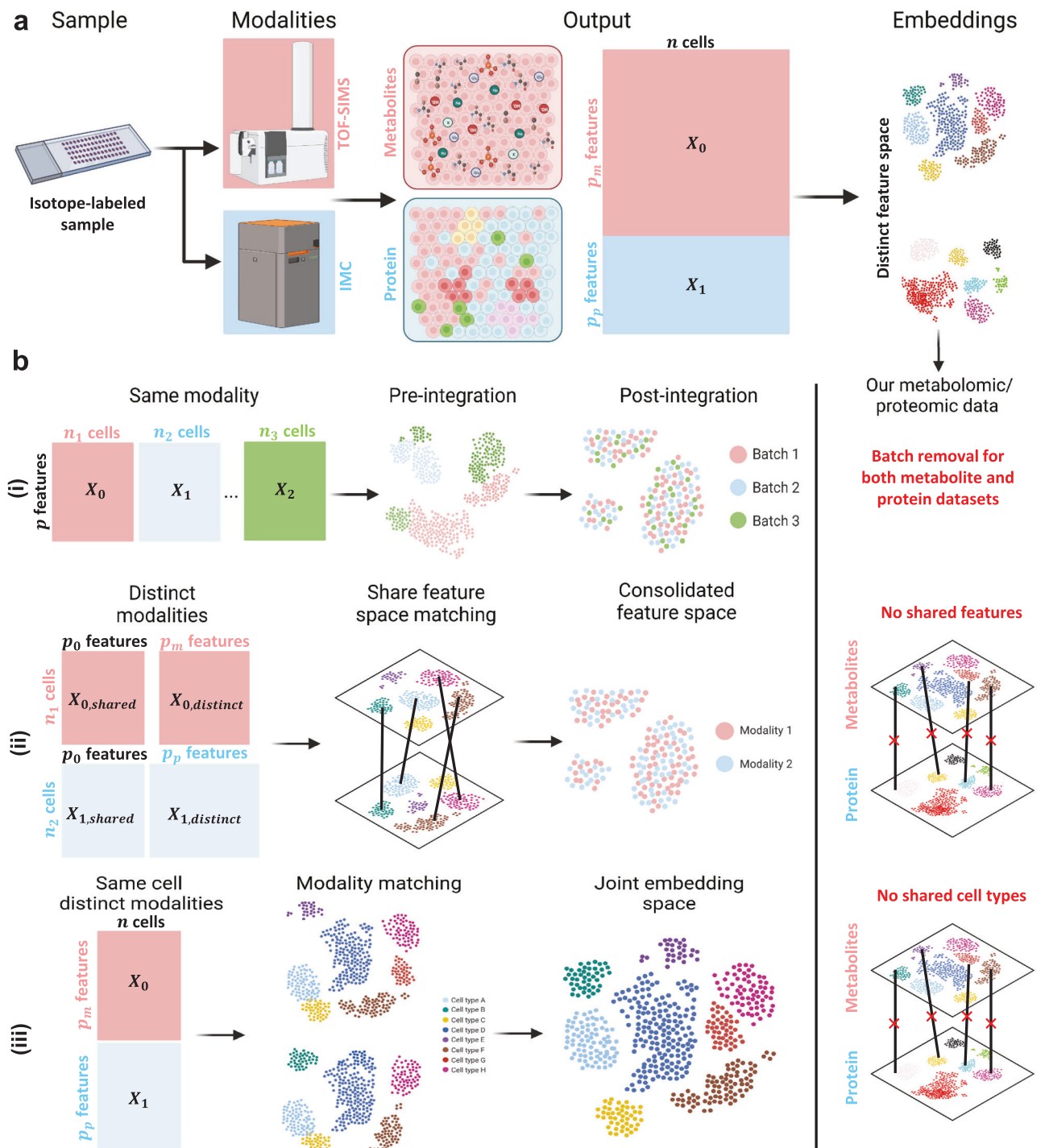

**Fig. 3 | Metabolomic and proteomic modalities need to be integrated with data analysis. a** Overview of metabolomic and proteomic data generated by scSpaMet. Tissue samples on glass slides are labeled with metal-isotope conjugated antibodies followed by metabolic profiling with TOF-SIMS and sequential proteomic profiling using IMC. The resulting outputs are $n$ cells with distinct $p_m$ metabolomic feature and $p_p$ proteomic features. The modalities have distinct feature spaces due to differences in feature number and variability across datasets. Created with Biorender.com. **b** Examples of existing spatial-omics data integration pipelines show (i) same-modality integration, (ii) cross-modality integration with shared features, and (iii) cross-modality multi-modal on the same cell. (i) is applied to remove the batch effect from both metabolite and protein datasets. Our scSpaMet data lacked any shared features (markers) or cell types. Created with Biorender.com.

adopted a cross-modality Variational Autoencoder (VAE) pipeline to integrate proteomics and metabolomic profiles from the same cell. Because of the imbalance of both data variability (i.e., low variation in metabolomic data compared to high variation in proteomic data) and size of the modality (i.e., 25 protein markers compared to more than 200 metabolite channels), VAE was used to detect specific metabolomic states of cell type subsets. Finally, in the multi-omics trajectory inference (3) we first used cell proteomics data to reconstruct cell differentiation trajectory in their spatial domain and correlated with metabolomic profile to extract the chemical pathways along cell trajectories. This sequential approach allows the study of metabolomic variation along cell trajectory defined by proteomic data.

Taking into account the large number of protein markers profiled using IMC, we applied the Leiden algorithm[37] for unsupervised clustering of single-cell proteomics data. The Leiden algorithm is widely used in unsupervised single-cell expression clustering[38]. After inspecting the resulting cluster and assigning it to the corresponding cell phenotype, we studied the correlation of cell type with metabolite channels in situ. Moreover, high-resolution in situ imaging data provided critical spatial information at the single-cell level and allowed cell-cell distance quantification. Currently, most of the spatial omics frameworks study the cell type neighboring frequency[30], probability[39], and cell-cell interaction[40] but not in cell competition from a metabolomic aspect. Therefore, we developed a single-cell multi-omics competition framework by comparing neighboring cell type metabolite ratios and as a function of distances to other cell types such as endothelial cells. Next, VAE[41] has shown an improved ability to model single-cell data distribution in both proteomics[27] and transcriptomics[25] with the ability to integrate different modalities and therefore was adopted here to extract joint latent representation from the proteomic and metabolomic data. Finally, single-cell trajectories have largely been limited to RNA-based technologies[42] due to higher throughput gene sequencing and proteomic trajectory analysis for well-defined cell lineages[43]. Here, single-cell trajectories were reconstructed with diffusion pseudotime analysis[29] using the previously established developmental progression of B-cells in human tonsil follicles characterized by their protein expression.

### Cell type-specific metabolic states in lung cancer tissues

Cancer cells can change their metabolic programming to meet their increasing energy needs for rapid progression and aggressive growth. This results in a tumor microenvironment (TME) depleted of critical nutrients, causing hypoxic and acidic conditions. These factors can lead to the lower recruitment and activation of the tumor-effector immune cells. It is critical to assess the differential metabolic requirements of the diverse immune cells in response to the growing tumors to uncover novel metabolic relations that can open the door for therapeutic interventions combining chemical perturbations with targeted immunotherapies[3].

We applied scSpaMet to a lung tumor microarray for the characterization of the protein-metabolic environment inside the TME (Supplementary Figs. 11–13). A tumor microarray of lung adenocarcinoma (grade III) with the tissue (TissueArray, ID: BS04081a) was stained with an antibody panel of 21 markers (Supplementary Table 3) spanning tumor, stromal, endothelial, and immune markers. The scSpaMet first identified single-cell protein phenotypes from the isotope-conjugated antibody panel of IMC using the Leiden algorithm to find the clusters of cells ($n = 10$) with similar expression profiles and annotated clusters by calculating their mean expression (Fig. 4a). Cancerous/paracancerous regions were labeled with clusters containing high expression level of pan-keratin and e-cadherin whereas stromal regions were marked by the expression of smooth muscle α-smooth muscle actin (α-SMA) and collagen type 1 (COL1). Immune cells were extracted based on the expression of specific protein markers (CD4, CD8, CD68, and CD3). The representative patients' tissue images were then reconstructed from the cell masks and their clustering results by assigning each segmented cell to its corresponding cluster (Fig. 4b and Supplementary Fig. 14). In each patient tissue image, single cells ($n = 19507$) were then classified into tumor and stroma regions based on their protein phenotypes and spatial localization enabling comparison of highly expressed metabolites at the single cell level in each of the two regions (Fig. 4c).

The scSpaMet captures the products of glycolysis metabolism instead of directly profiling glucose-associated small molecules due to inefficient detection by the TOF-SIMS instrument. Cancer cells upregulate the glycolytic catabolism of glucose into lactate even under normoxia. That leads to elevated levels of lactate, adenosine,

kynurenine, ornithine, reactive oxygen species (ROS), and potassium, contributing to the suppression of an anti-tumor response. Mass channels characterized by mass-to-charge ratios (M/z) were obtained from scSpaMet and correlated with known annotation from literature search and categorized the corresponding metabolite peaks into glucose, cholesterol, amino acid, and lipid fragments (See Methods). Selected mass channels of 74.0 m/z Glycine, 89.0 m/z Lactaid acids, and 122.0 m/z Adenine related to glucose metabolism have higher expression in tumor regions (Fig. 4d). On the other hand, Cholesterol fragmentation channels show higher cholesterol expression levels in stromal regions compared to tumor regions. Cholesterol is one of the most essential lipids for the cells' development, but cancer cells show more rapid depletion of the cholesterol than normal cells indicating their uncontrolled proliferation[44]. For identified amino acid and lipid-related channels, the single-cell expression levels inside stroma and tumor regions exhibit high variability (Fig. 4e). Single-cell metabolite spatial maps were reconstructed to visualize metabolic variation across regions in patients' tissue images. The 25 m/z lipid, 74 m/z Glycine, and 109 m/z Cholesterol fragments were shown in their spatial localization with correlation to define tumor and stromal regions (Fig. 4f).

### Metabolite competition of cell neighborhoods as a function of distance to vessels

Metabolic reprogramming occurs in tumor and non-tumor components of the TME by metabolic competition around tumor cells for a steady supply of nutrients even under hypoxic conditions[45]. Endothelial cells inside the lining of the vascular system play essential roles in the TME for promoting or preventing tumor progression[46] to support tumor metabolism[47] and metabolic reprogramming[4].

To study the impact of nutrient delivery around vascularization sites on chemical regulation, we developed a framework for single-cell local metabolite competition analysis in the lung TME. First, by leveraging CD31 protein markers from IMC multiplex data, we defined CD31+ endothelial cells in each patient tissue image. To consider the small size of the imaging field of view (FOV) in the 3D-SMF pipeline, we used the whole Tissue Micro Array (TMA) core IMC images to detect CD31-positive cells. Single-cell CD31 intensity expression followed a bi-modal Gaussian distribution and cells with higher intensities were defined as CD31+ endothelial cells and validated by inspecting original CD31 marker images (Supplementary Fig 15). We matched each 3D-SMF image region back into the IMC images (Supplementary Figs. 16 and 17). Then, for each segmented cell in the 3D-SMF image region, the minimum distance to CD31+ endothelial cells is extracted by identifying the cell centroids' position closest to the CD31+ endothelial cells from larger IMC-CD31 images by k-nearest neighbor algorithm (k-NN) search with spatial data of single-cells (Fig. 5a and Supplementary Figs. 16–18). In each tissue sample, the distances to CD31+ cells are used to generate metabolomic gradients (normalized in the range of 0 to 1) from distance maps by binning distance into 20 bins and averaging cell metabolite expression per bins (Fig. 5b). Specific lipid channels such as 25.0 m/z, 49.0 m/z, and 33.0 m/z are up-regulated around the CD31+ endothelial cells whereas 26.0 m/z, 74.0 m/z, and 98.0 m/z are up-regulated further away from CD31+ endothelial cells (Supplementary Fig. 19).

Neighboring cells in the TME enter a local nutrient competition due to the high proliferation nature of tumor cells and their need for nutrients. Here, we quantify the metabolite ratio of neighboring T- and tumor cells by modeling the local metabolite competition of tumor and CD3 + T-cells as a function of distance to CD31+ endothelial cells (Fig. 5c and Supplementary Fig. 20a). First, we construct a single-cell neighboring map from single-cell spatial data by specifying a radius of 20 μm. For each cell, we define the metabolite competition ratio as the metabolite expression of the cell divided by the median metabolite expression of its neighboring cells (i.e., a ratio of 1 would mean an equal metabolite level between a cell and its neighbors). Finally, we

combined the analysis of cell competition with distance to CD31+ endothelial cells. The mass channels of 148.0 m/z Methionine, 42.0 m/z lipid, 94.0 m/z, and 45 m/z lipid fragments have higher expression in tumor cells compared to T-cells at a further distance from CD31+ endothelial cells. On the other hand, the mass channels of 145.0 m/z Glutamine, 48.0 m/z, and 27.0 m/z exhibited higher expression in tumor cells compared to T-cells in the proximity of CD31+ endothelial cells. For T-cells, 148.0 m/z Methionine, 42.0, 81.0, and 100.0 m/z lipid

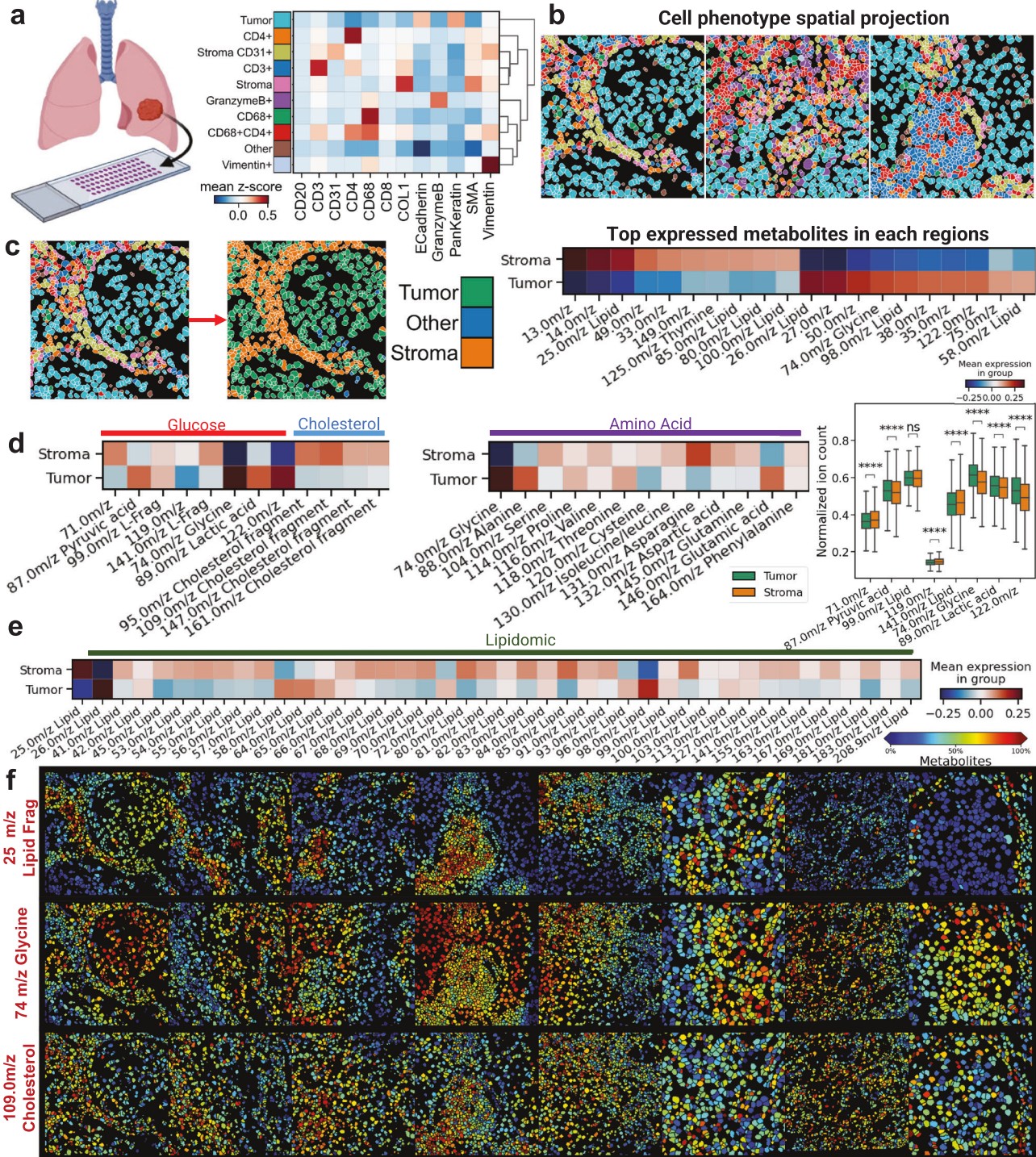

**Fig. 4 | The scSpaMet pipeline identifies metabolite differences between tumor and stromal regions in human lung cancer tissues. a** Unsupervised single cell clusters from protein profiles in human lung cancer tissues. Created with Biorender.com. **b** Spatial projection of corresponding cell clusters from a. **c** Left: definition of stroma and tumor region in tissue sample based on single cell phenotypes from (**a**). Right: The most expressed metabolite channels in stroma and tumor region from the differential analysis. **d** Comparison of single cell metabolite expression level for identified mass channels. Left: The metabolite channels related to Glucose pathway and Cholesterol fragments. Middle: The metabolite channels related to amino acid fragments. Right: Bar graph of selected metabolite channels (*n* = 19507 cells). Mann-Whitney-Wilcoxon test was two-sided with Bonferroni correction (ns: 0.05 < *p*, ****: *p* < =0.0001). All box plots with center lines showing the medians, boxes indicating the interquartile range, and whiskers indicating a maximum of 1.5 times the interquartile range beyond the box. **e** Metabolite channels related to identified lipid fragments. **f** Spatial projection of single cell metabolite expression level for selected metabolite channels.

fragments were upregulated near CD31+ cells. The scSpaMet reconstructed the metabolite competition maps of neighboring cells as a function of relative distances to the CD31+ endothelial cells overlaid on a spatial metabolomics map of the tissue.

Similarly, we analyzed the tumor and CD68+ cells' local metabolite competition as a distance from CD31+ endothelial cells (Fig. 5d Supplementary Fig. 20b). The mass channels of 129.0 m/z, 134.0 m/z Adenine, and 126.0 m/z Glycerophosphate demonstrated higher

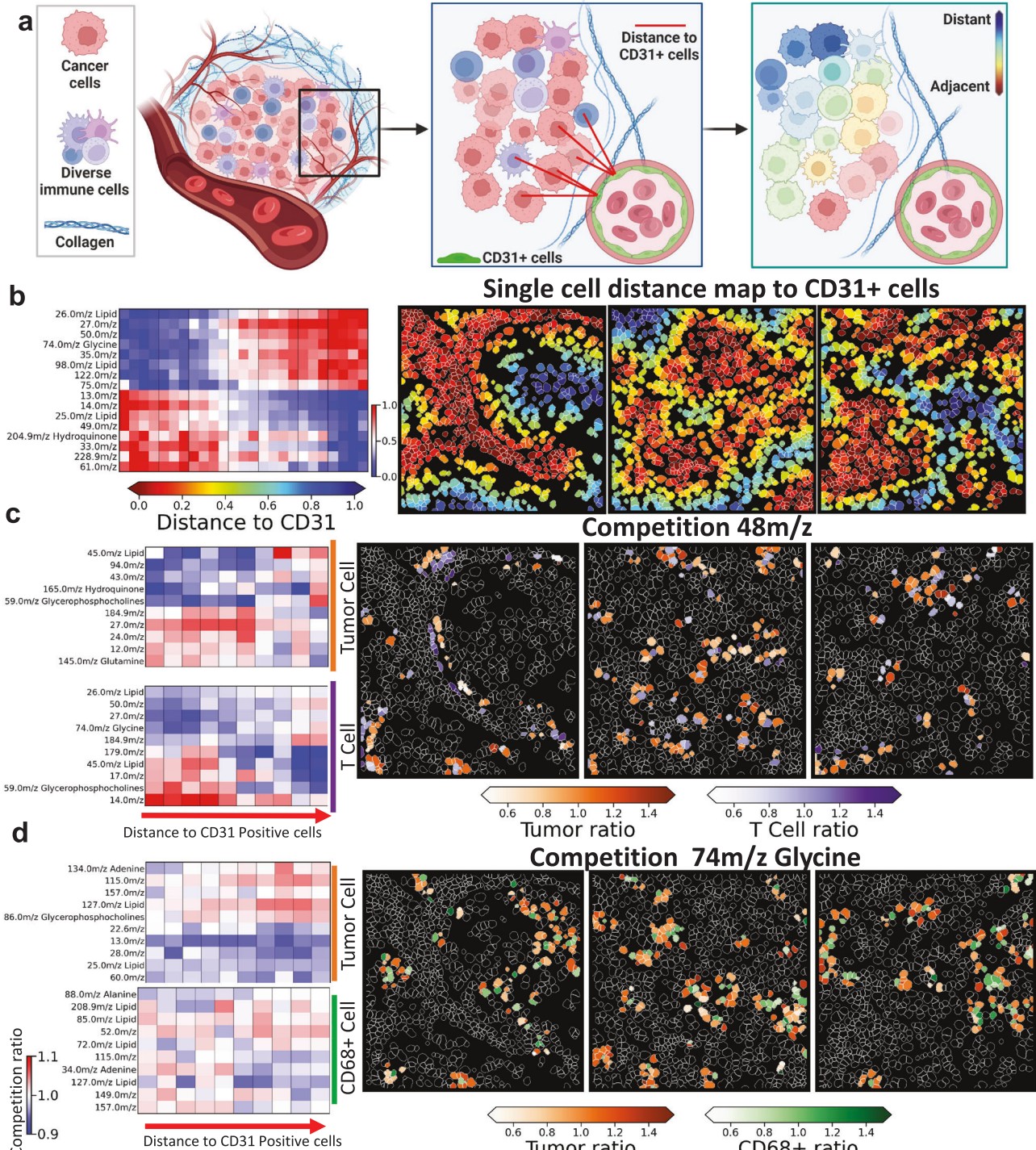

**Fig. 5 | The scSpaMet pipeline quantifies local metabolite competition in lung cancer as a function of distance to the endothelial cells. a** Representative schematic showing the definition of distance to CD31+ cells in lung cancer tissues. Created with Biorender.com. **b** Pearson correlation of metabolic signals compared to single cell distance to CD31+ cells. Left: Heatmap showing Pearson correlation of selected metabolite channels compared to the distance to CD31+ cells. Right: Single cell spatial distance map to CD31+ cells in lung cancer tissues. **c** Local metabolite competition as a distance of CD31+ cells between T-cells and tumor cells. Left: Selected metabolite channels showing positive and negative correlation of tumor cells (top) and T-cells (bottom). Right: spatial projection of T-cells and tumor cells' local metabolite competition for 48 m/z. **d** Local metabolite competition as a distance of CD31+ cells between CD68 positive cells and tumor cells. Left: Selected metabolite channels showing positive and negative correlation of tumor cells (top) and CD68 cells (bottom). Right: spatial projection of CD68 cells and tumor cells' local metabolite competition for 74 m/z.

expression in tumor cells compared to CD68+ cells at a further distance from CD31+ endothelial cells. On the other hand, the mass channels of 165.0 m/z Hydroquinone, 41.0, 42.0, 167.0 m/z lipid fragments, and 131 Asparagine exhibited higher expression in tumor cells compared to CD68+ cells in the proximity of CD31+ endothelial cells. For CD68+ cells, only the mass channel of 208.9 m/z lipid fragment was upregulated further from CD31+ cells. The mass channels of 61.0, 60.0, 77.0, and 28.0 m/z yielded high expression in CD68+ cells compared to tumor cells consistently. In the proximity of CD31+ endothelial cells, the mass channels of 134.0 m/z Adenine, 118.0 m/z Threonine, and 144.0 m/z were upregulated in CD68+ cells. The corresponding spatial map of local metabolite competition was visualized for cancer and CD68+ cells.

### Spatial metabolomic differences of lung tissues from distinct cancer patients

Lung cancer is heterogeneous not only at a cellular level[48] but also at the patient level with implications in the understanding of pathogenesis, diagnosis, and personalized therapy[49]. It is therefore imperative to decode the patient-to-patient variability of protein and metabolite distributions in single cells from different tumor biospecimens. The scSpaMet pipeline enabled the quantification and comparison of metabolite profiles at the patient level in lung cancer data. For each patient, scSpaMet was used to image 3 regions of interest per tumor. We then extracted the patient-level metabolite distribution and compared it across patients (Supplementary Fig. 21a).

Patient-level metabolite distributions were stratified into high-variation and low-variation metabolite channels (Supplementary Fig. 21b and Supplementary Table 4). By comparing patient-level metabolite distribution, lipid fragments were distributed between high and low-variation channels. Cholesterol fragment channels (95.0 m/z and 147.0 m/z) indicated low variation across patients. Glucose pathway-related channels such as 71.0 m/z, 87.0 m/z pyruvic acids, 89.0 m/z lactic acid, and 119.0 m/z demonstrated considerable variation across samples. Next, we quantified the variability of single-cell competition for tumor/T-cells and tumor/CD68+ cells for annotated metabolite channels (Supplementary Fig. 22). For T-cells and CD68+ cells, we plotted the mean competition expression for metabolite channels corresponding to glucose, cholesterol, amino acid, and lipid fragments.

To enable robust comparisons across metabolic and proteomic profiles of distinct patients, the scSpaMet analyzed joint protein-metabolite maps with a VAE architecture (Supplementary Figs. 21c and 23a and Methods). The VAE takes input from both the metabolite and protein profile of single cells and outputs the reconstructed profiles, that is a vector of size 21 for protein data and size 198 for metabolite data. The embedding layer of the VAE was used to capture a latent variable of the joint protein-metabolite profile at the single-cell level by combining a latent vector of size 8 from each modality. The resulting latent space embedding is used for clustering and showed multimodal phenotyping showing distinct metabolite states in tumor and stromal cell types (Supplementary Fig. 23b). By stratifying single-cell bi-modal spatial maps using self-supervised learning (Methods), we uncover the intra- and inter-patient variability at the analyzed region of interest level. Moreover, we decode the spatial joint metabolic-proteomic signatures for different groups of patients. To generate spatial signatures incorporating joint metabolomic and proteomics levels, we obtain unsupervised clustering labels for each cell based on their VAE latent embedding and we extract each cell neighboring information by taking a radius threshold of 25 $\mu$m. For each cell, we count the unsupervised cluster labels in its neighborhood and obtain a vector corresponding to the count of each clustering type around its neighborhood. After normalization of the vectors by their total count (i.e., with a density equal to one per cell), we perform a second unsupervised clustering using the spatial signature of joint metabolic and

proteomics profiles (Fig. 6a). We uncover the intra- and inter-patient spatial signature variability in each region of interest (Fig. 6c, d).

### Cell type-specific metabolic states around B cell follicles in tonsil tissues

Tonsils play an important role in the immune system and are part of the secondary lymphoid organs. They are composed predominantly of B- and T-cell populations in coordination with other immune cells and epithelial cells around the tonsil follicle regions[35,50,51]. We reasoned that deciphering spatially resolved cellular composition around B-cell follicles and how metabolic variations occur within B-cell subsets would be informative for decoding the role of chemical balance in humoral immunity. The scSpaMet was applied to healthy human tonsil tissues (Supplementary Table 5) to characterize the protein-metabolic environment around B-cell follicles (Supplementary Figs. 25–27). Herein, an antibody panel of 25 markers included immune surface markers, cytokine markers, epigenetic regulators, and extracellular matrix proteins (Supplementary Data 1). The scSpaMet first identified single-cell protein phenotypes ($n = 6$) using the Leiden algorithm (Fig. 7a) and representative patients' tissue images were then reconstructed from the cell masks, and their clustering results by assigning each segmented cell to its corresponding cluster (Fig. 7b and Supplementary Fig. 28a). In each patient tissue image, single-cell phenotypes ($n = 31156$) from unsupervised clustering were classified into the follicle zone, outside follicle zone, germinal center (GC) light zone (LZ) and GC dark zone (DZ) defining inside and outside GC regions, enabling comparison of highly expressed metabolites in each region (Fig. 7c and Supplementary Fig. 28b, c).

The inside and outside GC regions demonstrated statistically significant variations of metabolite distributions and the mass channels obtained from the scSpaMet were correlated with metabolite peaks annotated as glucose, cholesterol, amino acid, and lipid fragments. Rapid proliferation is key to affinity maturation, but little is known about how GC B cells fulfill the metabolic demands required to achieve the GC reactions. In lymphoid organs, B cells inside the GC undergo changes that lead to increased glucose consumption[52]. When B cells get stimulated through their B cell receptor (BCR) or the costimulatory protein CD40 or the Toll-like receptors (TLRs), the Hypoxia-inducible factor-1 (HIF-1) and the c-Myc expression are enhanced, leading to higher oxygen consumption, enhanced glycolysis, and increased production of lactate. This process causes higher consumption of amino acids including alanine and glutamine used as carbon and energy sources. Recent evidence suggested that GC B cells obtained the required energy from the fatty acid oxidation (FAO) pathway[53–55]. This finding was counterintuitive because other highly proliferative B cell blasts still exhibited high glycolysis activity, but GC B cells upregulated FAO while performing minimal glycolysis. Inside the GC regions, scSpaMet provided overall higher metabolite expression related to glucose fragments (71,.0, 87.0, 99.0, 119.0, 141.0 m/z), glucose pathway fragments (74.0 m/z Glycine, 89.0 m/z Lactic acids), cholesterol fragment channels, and amino acid-related channels but selected Fatty Acid channels (253.3 m/z and 277.0 m/z) showed higher expression outside of GC (Fig. 7c, d). Analysis of lipid-related channels demonstrated up-regulation in lipid fragments inside GC compared to outside GC (Fig. 7e). The mass channels of 25.0 m/z lipid fragment, 58.0 m/z lipid fragment, and 74 m/z Glycine fragment were shown in their spatial localization with correlation to distinct GC regions (Fig. 7f). We also used VAE architecture for joint protein-metabolite embedding. The result showed unique metabolic states inside and outside of GCs (Supplementary Fig. 29).

### Single-cell metabolite local competition around germinal centers

Humoral immunity against infections depends on the GC differentiation process in the B cell follicles of secondary lymphoid organs. In

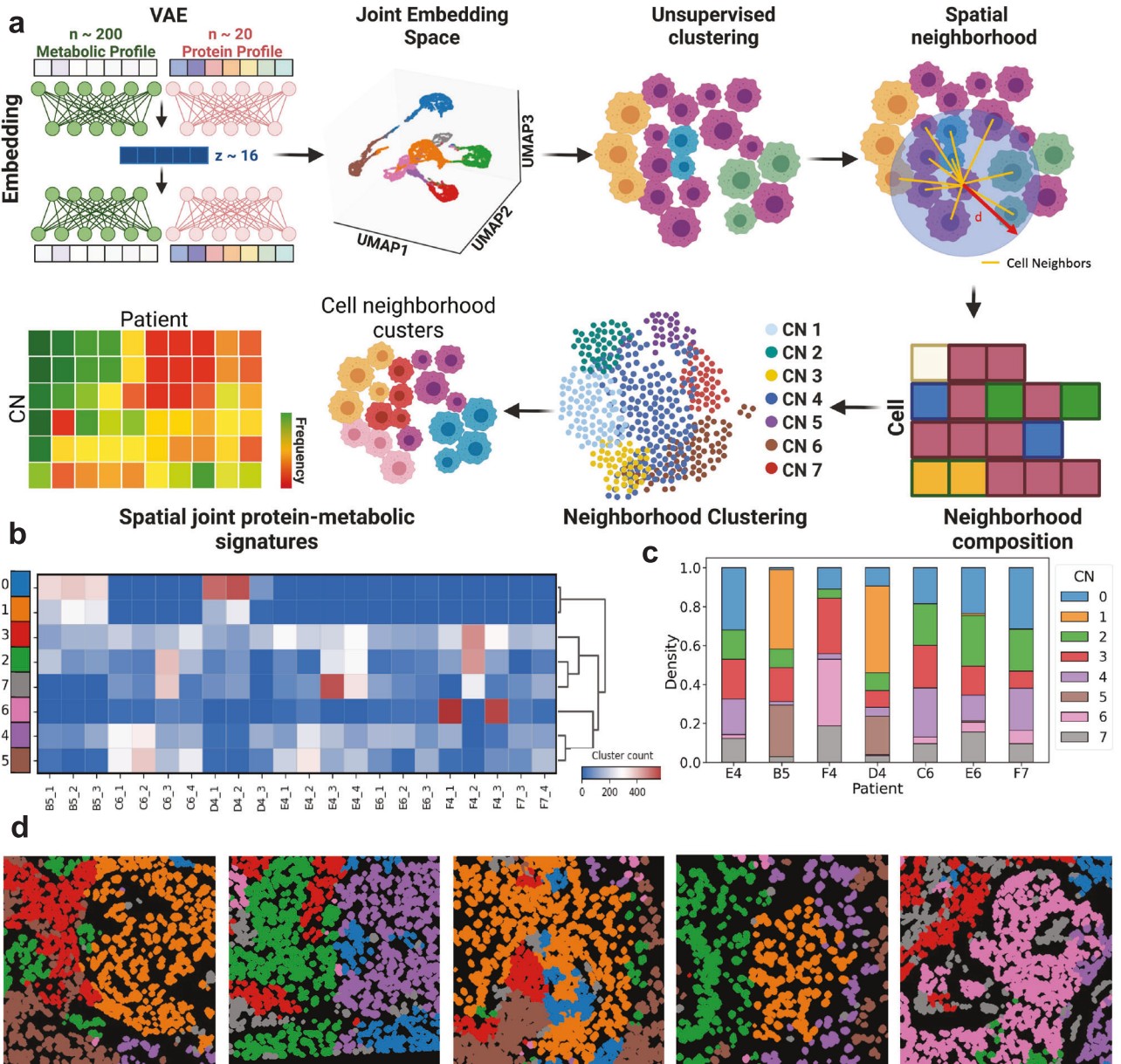

**Fig. 6 | The scSpaMet pipeline identifies spatial signatures of joint protein-metabolite signatures in lung cancer tissues across patients. a** Overview of spatial joint protein-metabolite signatures identification by the scSpaMet pipeline. Using VAE joint embedding, cells are clustered based on their joint protein-metabolite profiles. A neighborhood graph is constructed based on cell spatial location. The corresponding cell neighborhood cell type frequencies are used to determine spatial joint protein-metabolite signatures across patients. Created with Biorender.com. **b** Count of corresponding spatial joint protein-metabolite signatures across imaging regions in all the patients and their lung cancer tissues. **c** Frequency of corresponding spatial joint protein-metabolite signatures across patients in lung cancer tissues. **d** Spatial projection of cell corresponding joint protein-metabolite signatures from (**a**).

GCs, naïve B cells rapidly proliferate in response to T cell-dependent antigens and somatically mutate into high-affinity antibody-secreting cells, i.e., plasma cells[56]. In this GC process, B cells rapidly alternate between distinct "metabolic states" across quiescence, proliferation, and differentiation[57]. Rapid proliferation is key to affinity maturation, but little is known about how GC B cells fulfill the metabolic demands required to achieve the GC reactions. The scSpaMet enables the characterization of neighboring cell-to-cell local metabolite competition inside the GC. The phenotypes of single cells were identified inside the tonsil GCs using cell protein profiles to determine GC B-cells, T-cell follicular helper cells (TFHs), and Follicular dendritic cells (FDCs) with their corresponding metabolic distributions in their spatial environment. Similar to the lung cancer competition pipeline, local cell

competition of per-cell metabolite ratios was calculated as the metabolite expression of the cell divided by the average metabolite expression of its neighboring cells. (Fig. 8a, b and Supplementary Fig. 30). In human tonsil tissues, because of the high density of single cells in GCs, we defined local cells in competition when the single cell masks shared a boundary.

B-cell tonsil GCs are polarized into a LZ and DZ with functional and phenotypic distinction at the single-cell level[58]. Ki67 is a protein marker for the characterization of cell proliferation[59]. The scSpaMet identified GC LZ and DZ using Ki67, CD20, CD21, CD38, and EZH2 markers (Fig. 8c), and their metabolic distributions were compared. GC LZ exhibited a higher expression of 89.0 m/z lactic acid, 88.0 m/z Alanine, and several other amino acid channels.

## Metabolic trajectory analysis of pseudotime B cell differentiation

GCs of lymphoid organs are the place where activated B-cells undergo differentiation across DZ B-cells, LZ B cells, Memory B-cells, and Plasma cells[60–62]. DZ contains the rapidly dividing B-cells undergoing somatic hypermutation (SHM) and LZ contains FDCs, TFH, and B-cells that are exiting the GC area. B-cell migration happens inside GC from DZ to LZ[63]. Moreover, recent studies have provided ample

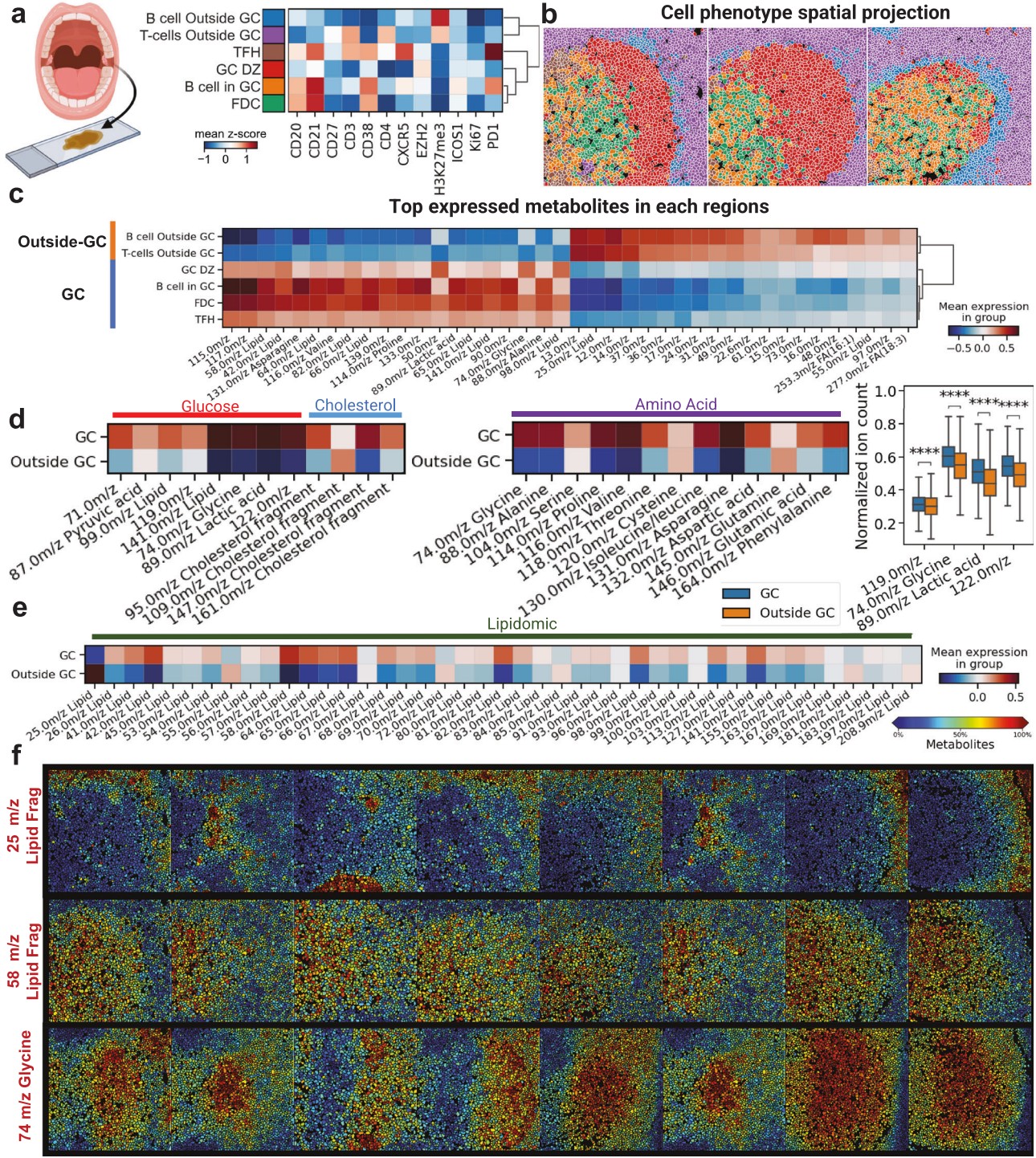

**Fig. 7 | The scSpaMet pipeline identifies metabolite differences of B cell follicles regions in human tonsil tissues. a** Unsupervised single cell clusters from protein profiles in human tonsil tissues. Created with Biorender.com. **b** Spatial projection of corresponding cell clusters from a). **c** The top expressed metabolite channels in regions inside and outside germinal centers across tonsil tissues from the differential analysis. **d** Comparison of single cell metabolite expression levels for identified mass channels. Left: The metabolite channels related to Glucose pathway and Cholesterol fragments. Middle: The metabolite channels related to amino acid fragments. Right: Bar graph of selected metabolite channels ($n = 31156$ cells). Mann-Whitney-Wilcoxon test was two-sided with Bonferroni correction (****: $p < =0.0001$). All box plots with center lines showing the medians, boxes indicating the interquartile range, and whiskers indicating a maximum of 1.5 times the inter-quartile range beyond the box. **e** Metabolite channels related to identified lipid fragments. **f** Spatial projection of single-cell metabolite expression levels for selected metabolite channels.

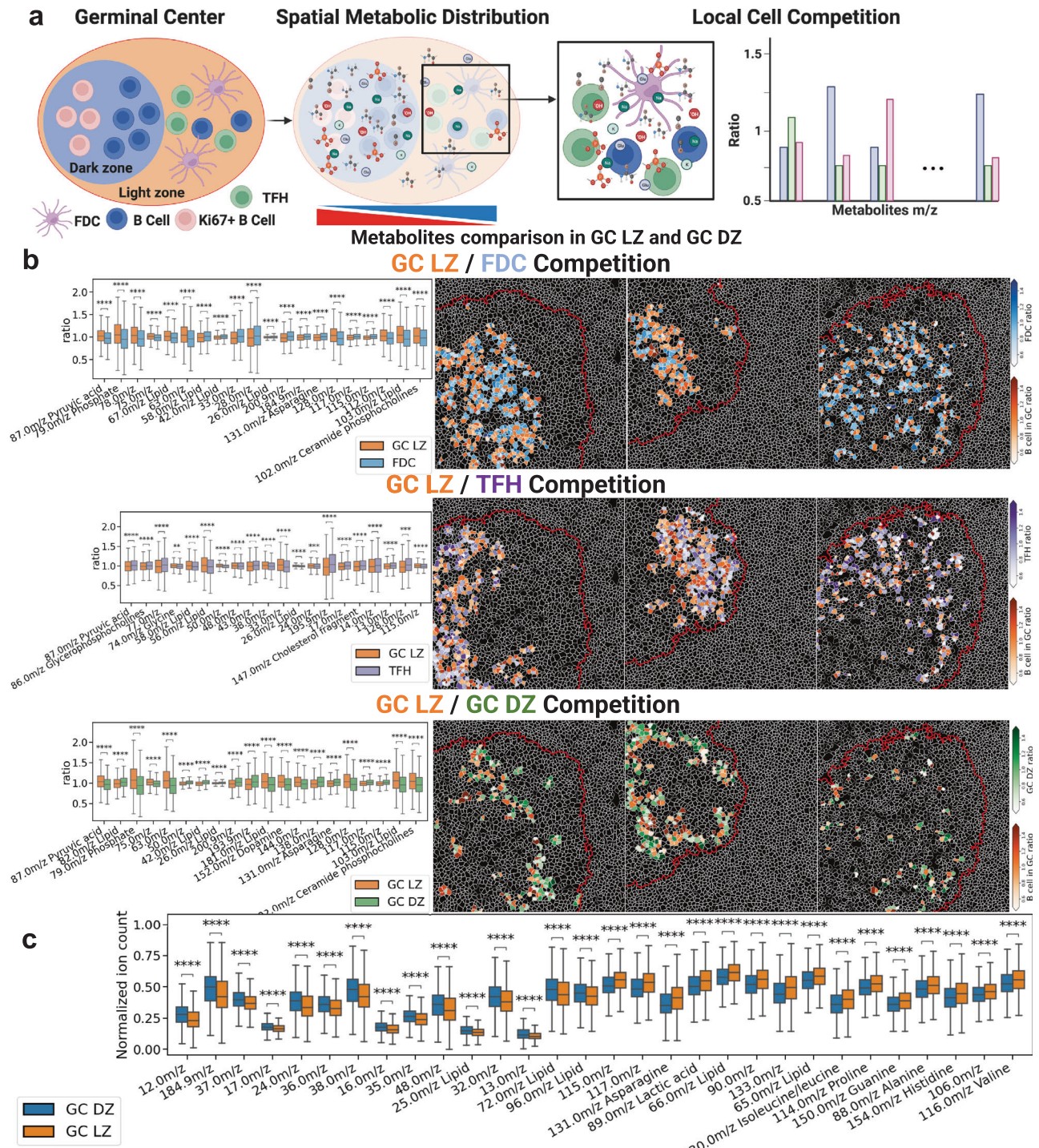

**Fig. 8 | ScSpaMet pipeline quantifies cell type-specific local metabolite competition in germinal centers. a** Representative schematic showing the definition of local cell metabolite competition in human tonsil germinal center regions. Created with Biorender.com. **b** Local competition of metabolites between B cells and FDCs (top, *n* = 4371 cells), B cells and TFHs (middle, *n* = 2807 cells), and B cells in LZ with DZ (bottom, *n* = 1870 cells). Mann-Whitney-Wilcoxon test was two-sided with Bonferroni correction (ns: 0.05 <p, ****: *p* < =0.0001). All box plots with center

lines showing the medians, boxes indicating the interquartile range, and whiskers indicating a maximum of 1.5 times the interquartile range beyond the box. **c** Comparison of selected metabolite channels between GC LZ and GZ DZ. Mann-Whitney-Wilcoxon test was two-sided with Bonferroni correction (ns: 0.05 < *p*, ****: *p* < =0.0001). All box plots with center lines showing the medians, boxes indicating the interquartile range, and whiskers indicating a maximum of 1.5 times the interquartile range beyond the box.

experimental evidence for the re-entry of selected B cells from LZ to DZ upon antigen-driven selection[64–66]. B-cell pseudotime analysis inside germinal centers has been used to trace their developmental trajectories using RNA-seq and protein data[67–70]. The scSpaMet enables B-cell trajectory analysis from single-cell protein expression with metabolite correlation in their spatially resolved tissue coordinates. By

incorporating the single-cell protein phenotypes inside tonsil GC, the scSpaMet infers the pseudotime trajectory of the GC B-cells (Fig. 9a).

Using multiplexed protein markers, B-cells inside of GC (*n* = 15655) were selected for unsupervised phenotyping with the Leiden algorithm (Supplementary Fig. 31a), projected into their t-distributed stochastic neighbor (t-SNE) embedding space, and their

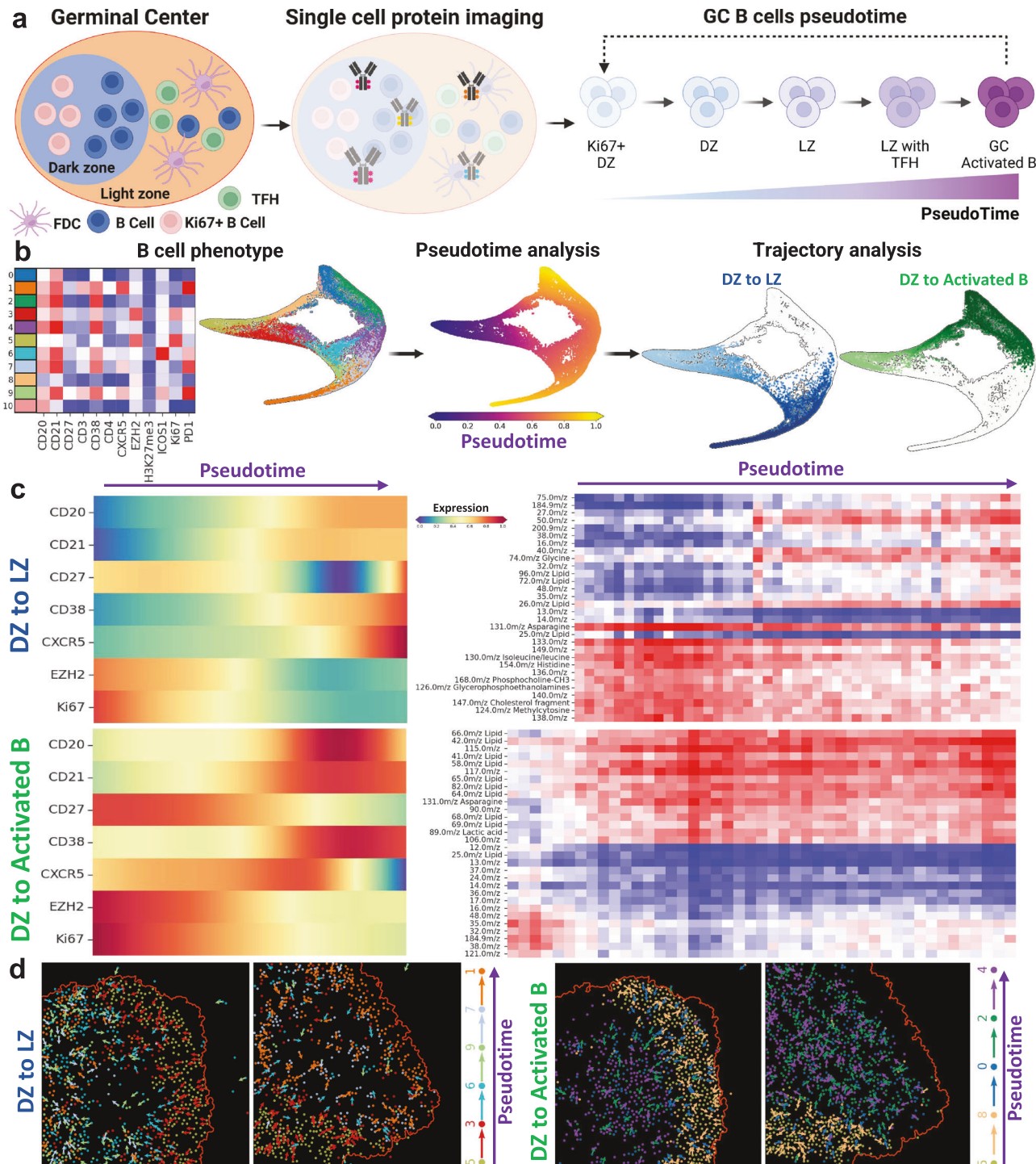

**Fig. 9 | The scSpaMet pipeline infers metabolites trajectory for B cell differentiation inside of germinal centers. a** Representative schematic showing the definition of germinal center B cell trajectories. Created with Biorender.com. **b** B cell trajectories in the germinal center from protein markers in scSpaMet. Left: Unsupervised clustering of B cell protein markers and corresponding TSNE plot. Middle: Pseudotime analysis of B cell protein phenotype. Right: Identified B cell trajectories from single-cell phenotype. (1) the GC DZ B-cells to GC LZ B-cells and (2) the GC DZ B-cells to the activated B-cells. **c** Identified germinal center DZ to LZ and DZ to Activated B-cell trajectory. Left: Variation of protein marker intensity along identified trajectories, right: Variation of selected metabolite channel along identified trajectories. **d** Spatial plot of germinal center DZ to LZ trajectory (left) and germinal center dark zone to Activated B cell trajectory (right).

pseudotime ordering was inferred by determining the cell state and calculating the respective probability of differentiating into each terminal state[71] (Supplementary Fig. 31b, c). This process computationally reconstructed two different differentiation trajectories of B cells inside GC (Fig. 9b). Here we infer single-cell hierarchy from their protein profiles. We selected the starting point from single-cell

expression corresponding to DZ B-cells. After plotting the cells in the embedding space (Fig. 9b), we determine two distinct trajectories that represented DZ to LZ and DZ to activated B cells from single-cell protein expression across trajectories and their corresponding value for the diffusion pseudotime. The bifurcation point was determined empirically by the embedding plot of single cells. The scSpaMet then

characterized the metabolic variations along the defined B-cell trajectories, including the pseudotime path (1) the GC DZ B-cells to GC LZ B-cells and (2) the GC DZ B-cells to the activated B-cells demonstrating higher CD38 expression (Fig. 9c).

To visualize the gradients of B-cell trajectory in the spatial domain, we projected defined B-cell phenotypes from trajectory analysis onto their spatial domain. Each cell is represented by a scatter point with the color corresponding to its cluster information. Following the identified trajectory paths for each cell in a cluster along a path, we define the spatial direction by taking the five nearest neighbors of the cell in the next cluster on the path and plot an arrow to the centroids of the nearest neighbors (Fig. 9d and Supplementary Fig. 32). This allows the spatial reconstruction of B-cell differentiation trajectory inside of the GC in human tonsil tissues. Similarly, the corresponding states of single-cell differentiation were projected back into their t-SNE embedding space for visualization and represented as a graph-directed method (Supplementary Fig. 33). On the other hand, we projected the values of single-cell pseudotime into their spatial domain and the corresponding pseudotime trajectory gradient to visualize the spatial spread of these trajectories (Supplementary Figs. 34 and 35).

## Spatial metabolomic profiling in endometrium tissues

Human endometrium, the mucous membrane lining the uterus, undergoes dynamic changes through remodeling, shedding, and regeneration during the menstrual cycle[72]. The temporal and spatial dynamics of endometrium cells have been studied at the single-cell level to dissect the signaling pathways that determine the cell fate of the epithelial lineages in the luminal and glandular microenvironments[73]. Studies have shown that increased body mass index is most strongly associated with endometrial cancer incidence and mortality[74,75]. It is therefore important to characterize the metabolomic variation between cell types and across conditions to better dissect the molecular mechanisms underlying how obesity contributes to endometrial cancer.

The scSpaMet was applied to human endometrium tissues for characterization of their protein-metabolic environment (Supplementary Fig. 36). Herein, an antibody panel of 9 markers comprised of immune surface markers, epithelial markers, and extracellular matrix proteins (Supplementary Table 6). The scSpaMet first identified single-cell protein phenotypes ($n = 4$) using the Leiden algorithm (Fig. 10a) and representative patients' tissue images were then reconstructed from the cell masks, and their clustering results by assigning each segmented cell to its corresponding cluster (Fig. 10b). In multiplexed images of each patient tissue, we extracted single-cell protein profiles ($n = 8125$) to characterize cell phenotype and conducted a comparison of highly expressed metabolites for cell type (Fig. 10d, e). Single-cell metabolite spatial maps were reconstructed to visualize metabolic variations across regions in lean and obese benign patients' tissues (Fig. 10f). Next, we quantified the metabolomic variation between lean and obese benign patient samples by comparing the annotated mass channel associated with glucose, cholesterol, amino acids, and lipids (Supplementary Fig. 37). Glucose pathway-related channels such as 74.0 m/z Glycine and 89.0 m/z Lactic acids showed high expression in obese benign samples whereas other glucose fragments (71.0 m/z, 87.9 m/z Pyruvic acids, 99.0 m/z) showed higher expression in the lean benign patient.

## Discussion

In this study, we developed ScSpaMet, a framework for joint protein-metabolite imaging at the single-cell level. The scSpaMet framework allowed systematic single-cell segmentation, phenotyping from protein data, and metabolite profiling on the same tissues at high spatial submicron resolution. While 3D-SMF captured incomplete molecular fragments[76], it produced highly multiplex metabolite

imaging and minimal sample alteration allowing further correlative imaging with IMC. This study shows the segmentation and analysis of single-cell protein and metabolic feature profiles directly in the same tissue section. We applied scSpaMet to human lung cancer, tonsil, and endometrium tissues. The scSpaMet identified metabolic variation between cells in the tumor and stromal regions, cell type-specific local metabolic competition, metabolic trajectories, and patient-level molecule variation in lung cancer tissues (Supplementary Fig. 38a). Similarly, scSpaMet quantified metabolic changes around B-cell follicles in human tonsil tissues by analyzing B-cells, T-cells, and FDCs inside GC LZ/DZ regions (Supplementary Fig. 38b), cell type-specific local metabolic competition inside of GC, and metabolomic changes along B-cell differentiation trajectories. Finally, we profiled human endometrium tissues using scSpaMet to decipher the metabolic composition of cell types from lean and obese benign conditions.

One potential limitation of this study is the challenge in metabolite annotation and coverage for lipids, amino acids, and the metabolic pathways from TOF-SIMS. Another limitation is the relatively small imaging regions from tissue samples. This is a trade-off between imaging speed and spatial resolution in the TOF-SIMS. However, multiplexed protein images from IMC were able to discern the cell type heterogeneity of the TME. The immune panel can be expanded to analyze further phenotypes such as macrophage M1/M2 or B-cells subtypes. Optimal cutting temperature (OCT)-embedded frozen tissues also showed better metabolite preservation compared to formalin-fixed paraffin-embedded (FFPE) samples, but the tissue structures were less preserved therefore the need to incorporate cryo-TOF-SIMS imaging into the pipeline (Supplementary Figs. 39 and 40). Finally, another aspect of this study was the lack of information related to the treatment history of the patients as it can alter the metabolite state within the tumors and help explain metabolic variation across patients.

Despite these limitations, the scSpaMet provided a complementary solution to the need for simultaneous whole-cell metabolic and protein analysis in situ by incorporating untargeted spatial metabolomics and targeted multiplexed protein imaging in a single pipeline. The scSpaMet registered single-cell measurements in bi-modality and enabled accurate identification of various cell types with their corresponding metabolomic profiles.

In summary, scSpaMet allows high-resolution joint protein and metabolite profiling at the single-cell level in the same tissue. Using protein markers, single cells are annotated, and their metabolic variation is quantified. Combining cell type spatial information and metabolic profile, a local cell metabolite competition framework is proposed. Moreover, metabolic reprogramming along cell differentiation trajectories can be retraced and projected spatially. With the advancement of spatial mass spectrometry imaging resolution, molecule annotation capabilities, and efficiency, scSpaMet paves the way to systematic single-cell metabolite and protein profiling in their tissue environment.

## Methods

### Ethics statement

Lung cancer and tonsil tissues were obtained from TissueMicroarray (Previously: BioMax) vendor where they were collected with patients' consent following high ethical and medical standards with the donor being informed completely and with their consent. All human tissues are collected under HIPPA-approved protocols. Due to the limited samples available, sex and gender were not considered in the study design. Endometrium tissues were obtained from patients with informed consent undergoing surgical resection procedures at Northwell Health Long Island Jewish Medical Center and shipped to Cold Spring Harbor Laboratory for processing (Collected by S. B.). Study protocols were reviewed and approved by the Northwell Health

Biospecimen Repository (Protocol number: NHBR 18-0897). Tissue samples were kept in RPMI medium (Cat#, 10-040-CV, Corning) until processing. The study was conducted following the criteria set by the Declaration of Helsinki.

**Tissue preparation and isotope-conjugated antibody labeling**
Patients' samples for lung tumors were obtained from a TMA purchased from a third-party vendor (TissueArray.com, US) with the tissue ID: BSO4081a. This TMA included a total of 63 tissue cores of FFPE non-

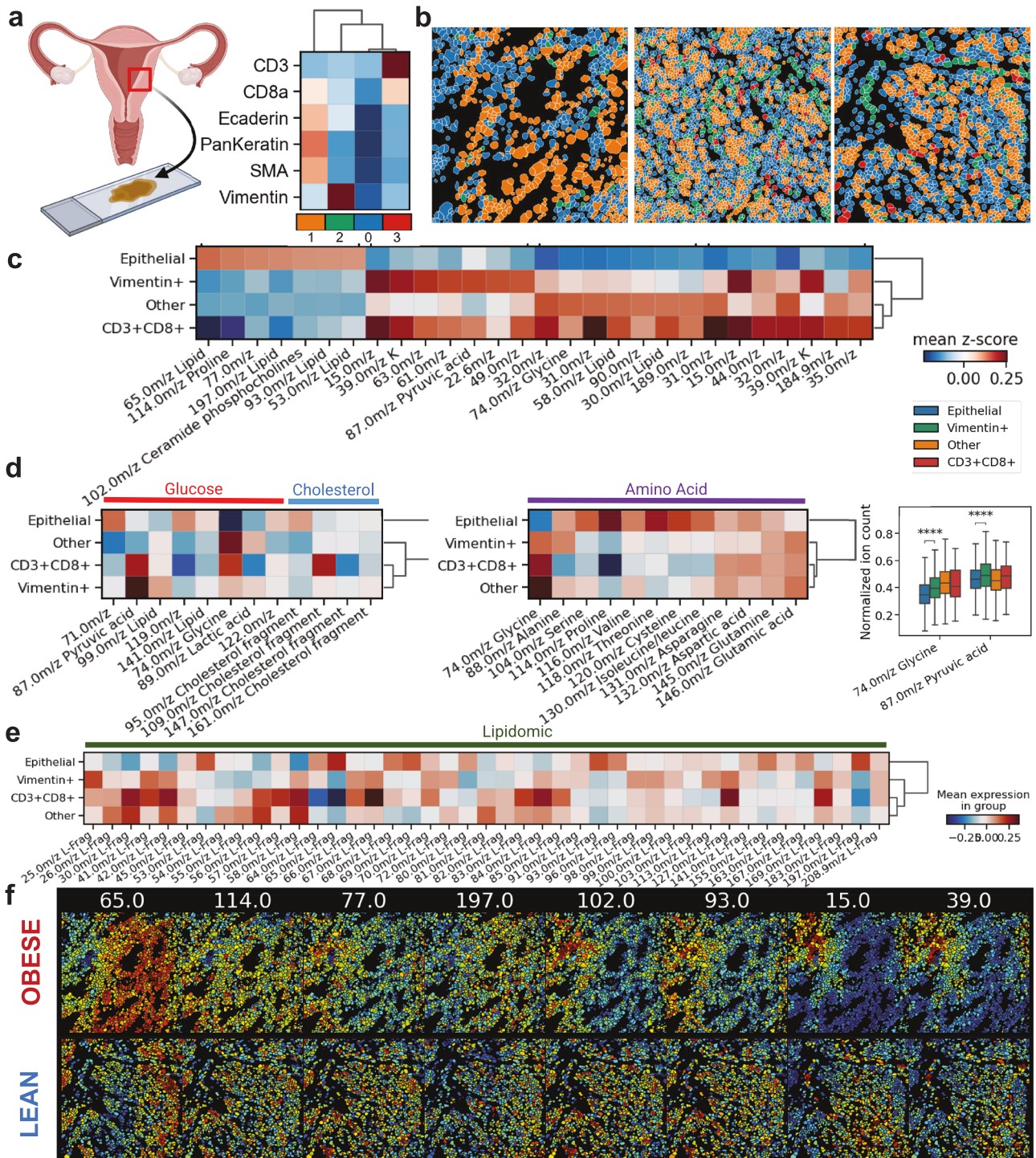

**Fig. 10 | The scSpaMet pipeline characterizes proteins and metabolites in human endometrium samples. a** Unsupervised single cell clusters from protein profiles in human endometrium tissues. Created with Biorender.com. **b** Spatial projection of corresponding cell clusters from **a**. **c** The top expressed metabolite channels in identified cell types across endometrium tissues from the differential analysis. **d** Comparison of single cell metabolite expression levels for identified mass channels. Left: The metabolite channels related to Glucose pathway and Cholesterol fragments. Middle: The metabolite channels related to amino acid

fragments. Right: Bar graph of selected metabolite channels (*n* = 8215 cells). Mann-Whitney-Wilcoxon test was two-sided with Bonferroni correction (****: *p* < =0.0001). All box plots with center lines showing the medians, boxes indicating the interquartile range, and whiskers indicating a maximum of 1.5 times the interquartile range beyond the box. **e** Metabolite channels related to identified lipid channels fragmentation. **f** Spatial projection of single-cell metabolite expression levels for selected metabolite channels in obese benign (top) and lean benign (bottom) tissues.

small cell lung adenocarcinoma and adjacent normal lung tissue samples obtained from 7 patients. We profiled 21 regions of interest from 7 cores. Each tissue core had a diameter of 1 mm and a thickness of 5-μm which is within the tissue thickness recommended for IMC ($\leq 7$-μm). The tissue labeling protocol was followed as previously reported in the protocol[35], including antigen retrieval, protein blocking, metal-tagged antibody labeling, and nucleus counterstains. The protocol starts with baking the FFPE sections at 60 °C for 2 hours to ensure that the tissues adhere to the glass slides. The samples were then dewaxed by immersing the slides into xylene and hydrated by sequential immersion in descending concentrations of ethanol (100%, 95%, 80%, 70%, and 50%). This step was then followed by several washes in Maxpar water (Catalog number 201069, Standard BioTools) where they were left to incubate for 5 minutes before proceeding with the antigen retrieval step. The heat-induced epitope retrieval method was used under basic conditions (pH =9) using Dako's target retrieval solution (Catalog number: S2367, Agilent Dako). A pressure cooker was used and set to high temperature for 15 minutes. The samples were then left to reach room temperature while immersed in the antigen retrieval buffer. The slides were washed with Maxpar Water and Maxpar PBS (Catalog number 201058, Fluidigm). A PAP pen was then used to draw a hydrophobic barrier around the tissues and Dako's ready-to-use protein-blocking buffer solution (Catalog number: X090930-2, Agilent Dako) was used to avoid non-specific binding for the antibodies. The antibody cocktail mix was prepared in the protein-blocking buffer and was left to incubate with the samples overnight at 4 °C. The next day, the samples were washed with 0.2% Triton X-100 in Maxpar PBS with gentle agitation. The samples were then stained with Intercalator-191Ir/193Ir prepared in Maxpar PBS for 30 minutes at room temperature. After the staining process was complete, the stained tissues were stored at 4°C until imaging time. The human tonsil tissue sections were from TissueArray.com with the tissue group ID: HuFPT161. Tonsil sample 1 had tissue ID SU1 and tonsil sample 2 had tissue ID SM2. Tonsil sample 1 had 5 imaging regions of interest and tonsil sample 2 had 6 imaging regions of interest.

## IMC imaging

To set up the Hyperion imaging system, regions of interest of 1500-μm x1500-μm were chosen within each tissue core to cover most of the tissue. To choose the most optimum laser ablation power, several testing points were chosen from the tissue cores that represent the tissue heterogeneity. The acquired data is automatically saved in.MCD format that can be viewed using Fluidigm's MCD Viewer software (v 1.0.560.2).

## IMC image processing

Each region of interest image was extracted using MCD Viewer (v 1.0.560.2) with a minimum threshold intensity of 0 and maximum threshold intensity of 50. Each image intensity range was then scaled to 0.1 and 99.9th intensity percentile for processing. Noise removal using a k-NN filter[77] is applied to reduce noise in the dataset.

## TOF-SIMS imaging

The TOF-SIMS (IONTOF 5 GmbH, Münster, Germany) instrument uses a bismuth liquid metal ion gun as the primary ion source to generate secondary ions from the sample surface, followed by identification of m/z of secondary ions by a TOF analyzer. The bismuth source can be used in different modes: Singly charged 25-kV monoatomic (Bi + ), singly charged 25-kV three-atom cluster (Bi3 + ), or doubly charged 50-kV cluster (Bi3 + +) mode. Distinct modes exhibit extra cluster mass and energy that can increase the yield (i.e., the number of secondary ions per primary ion) of heavier molecular weight secondary ion species. The Bi3 + + mode was chosen based on single-cell features (Supplementary Fig. 41a). Regions of 300-μm x300-μm to 400-μm x400-μm area were raster scanned at 512×512 pixels. Pixel densities were

chosen empirically (Supplementary Fig. 41b). The bright field images were used to find imaging regions identified in H&E sequential tissues (Supplementary Figs. 42 and 45). In this device, the secondary ions were collected and then accelerated across a voltage gap and directed into an ultrahigh-vacuum flight tube toward the detector (a combination channel plate and photomultiplier tube). The flight time of secondary ions is proportional to the ion m/z; thus, lighter ions (lower m/z) arrive at the detector more quickly, and the heavier the ion, the longer the TOF. For depth profiling, another cesium ion gun (Cs+ ions, 2-kV energy, and microampere current) was used to iteratively sputter away very thin layers, followed by bismuth (Bi) bombardment to generate secondary ions from the tissue sample before and after Cs sputtering cycles. Depth profiling allowed the detection of more molecules in imaging regions (Supplementary Fig. 46). Depth profiling in the TOF-SIMS was performed around 30-40 slices for 10 s ablation at 2-kV per slice for an estimated ablation of 1 micron per hour. In the 3D-SMF platform, negative and positive modes were experimentally tested. As the metabolomic profiling measurements in unlabeled samples provided more noteworthy single-cell spatial distributions in the negative mode, the rest of the experiments were performed to detect the negatively charged compounds as the selection of polarity (Supplementary Fig. 47).

## SIMS data preprocessing

The IONTOF SurfaceLab software (version 6) was used to perform basic image processing operations on the acquired spatial mass spectra. The spatial distribution for each selected peak was exported in files containing the coordinates and pixel intensity values. Around 200 peaks were selected using Surface Lab. The data were then exported in American Standard Code for Information Interchange (ASCII) format into a text file. For putative annotation of the metabolite channels, we used published work of TOF-SIMS metabolite imaging literature (Supplementary Data 2 and 3). For SIMS data, the mean ion count normalization was applied for intra-sample normalization[78]. Each pixel in the imaging data is normalized by their mean intensity over all extracted ion channels. If the pixel $i$ in the imaging data is represented by a vector $X_i = (x_{i,1},..., x_{i,p}) \in R^p$, the normalization is defined as

$$Y_i = \frac{X_i}{\sum_{j=i}^{p} x_{ij}} P \tag{1}$$

## Pixel clustering

Pixels in each imaged region of interest are concatenated and were downsampled by 20 folds to extract a subsampling of the whole dataset. The selected pixels were normalized to a mean of 0 and a standard deviation of 1. Then, a Leiden clustering algorithm was applied to the down-sampled dataset to assign a cluster label to each pixel. Finally, pixels of the entire dataset were assigned to the cluster of most likely neighbors among the 30 nearest neighbors in pixel feature space in the down-sampled dataset.

## Image registration

A two-step image registration was performed to match and align cross-modality images (SIMS, IMC, and H&E). First using a Fast Fourier transform (FFT) algorithm the cross-correlation between two images is calculated to get translational offsets. The position of the maximal correlation coefficient was identified and used to match images. After finding a translational offset and matching region, the rotation offset between the two aligned and cropped images is calculated using FFT on the polar space transformed of the images. Finally, the affine transformation of the two images is found by applying a Difference of Gaussian (DoG) filter local maxima above a selected threshold were selected as features and matched with the RANSAC algorithm[79]. To match the pixel density of the two modalities, we downsampled the

higher-density SIMS modality using bi-quadratic interpolation without anti-aliasing (Supplementary Figs. 48 and 50).

## Single-cell segmentation

In cancer tissues, single-cell nuclei regions were also segmented using the deep learning model Cellpose[80] by using Histone H3 marker in lung cancer samples, combining DNA1, DNA2, Ki67, and PD1 markers in tonsil samples and combining DNA1, DNA2 markers in endometrium samples from IMC modality. The cytosol region was calculated by expanding the nuclei-segmented region by 2 pixels. In tonsil tissues, we use the best available nuclei marker (Histone H3 in tonsil data) and we combine multiple protein markers using maximum projection to get the whole cell area (Selected for the tonsil data: CD38, Vimentin, CD21, BCL6, ICOS1, CD11b, CD86, CXCR4, CD11c, FoxP3, CD4, CD138, CXCR5, CD20, CD8, C-Myc, PD1, CD83, Ki67, COL1, CD3, CD27, and EZH2). To get the cytosolic region of each cell, we subtract the nuclei image from the combined protein marker image. Finally, we use the deepcell[81] Mesmer model for single-cell nuclei and cytosol segmentation. This 2-step segmentation pipeline better captures the overall cytosolic region of single cells by incorporating the majority of available multiplex imaging panels. The combined multiplex images were used for single-cell cytosol and nuclei segmentation. We compared it to a segmentation using a registered PO3- channel image from 3D-SMF imaging which provided nuclei signals in cells. We showed that the average captured nuclei area using the IMC modality yields higher cell areas compared to 3D-SMF and 2-fold higher in IMC cytosol segmentation compared to SIMS nuclei (Supplementary Figs. 51 and 52).

## Clustering algorithm

Single-cell unsupervised clustering was performed using the Leiden algorithm[37] a graph-based community detection algorithm. From each segmented cell region, the mean intensity of each marker expression was calculated. The resulting feature matrix consisted of n rows of the total number of cells ($n = 19507$ for lung, $n = 31156$ for tonsil, $n = 8215$ for endometrium) and p columns of marker expression. Each column of the feature matrix was normalized z-score and batch correction between samples was performed using Scanorama pipeline[20]. The neighborhood graph in the embedding space was constructed and used for unsupervised community detection. Each cell was associated with a cell phenotype cluster and attributed a cluster color showing the cell-level clustering of each region of interest.

## Spatial Neighboring map

From single-cell segmentation and clustering, cell centroids, and corresponding clusters were extracted. Spatial Neighboring maps were created by connecting centroids within 20-μm of each other in lung cancer and touching cell masks in tonsil samples, thus generating a spatial proximity network for each region of interest with corresponding cell phenotype. The average minimum and maximum axis length of single cells were used to infer a mean diameter of 10 μm and a distance of 20 μm was chosen based on 1 cell distance. Cell contacts are extracted by binary dilation of single-cell masks and cells that overlap were considered to have contact[31]. We choose cell mask contact for spatial neighboring maps generation in tonsil samples because of the higher density of cells.

## Local cell competition

Cell local metabolite competition is defined by cell spatial neighboring map. For each cell, we define the metabolite competition ratio as the metabolite expression of the cell divided by the average metabolite expression of its neighboring cells. This gives a metabolite competition ratio equal to 1 when the metabolite of a cell is equal to the average of its neighboring cells, a competition ratio less than 1 when the metabolite of a cell is less than the average of its neighboring cells, and a competition ratio greater than 1 when the metabolite of a cell is

greater than the average of its neighboring cells. Let $m_i$ be the metabolite expression level of cell $i$ and $m_{Ni}$ the average metabolite expression level of cell $i$ neighbors, the cell metabolite competition ratio is defined as: $\frac{m_i}{m_{Ni}}$.

## Variational autoencoder embedding

For protein and metabolites data integration at the single cell level, a Variational autoencoder (VAE) was used for embedding extraction[41]. The VAE has input x for single-cell, $f_E$, and $f_D$ represent the transformation by encoder and decoder layers. In addition to the standard autoencoder, two transformations $f_\mu$ and $f_\sigma$ are added to the output $e$ of the encoder to generate the parameters $\mu$ and $\sigma$ ($\mu, \sigma \in R^m$). The compressed data $z$ is now sampled from the normal distribution *with mean μ* and standard deviation $\sigma$. In contrast to the standard autoencoder, VAE uses $z$ as the input of the decoder instead of $e$. By adding randomness in generating $z$, VAE prevents overfitting by avoiding mapping the original data to the compressed space without learning a generalized representation of data. Formally given an input dataset x we want to infer the characteristics of z by learning the posterior distribution $p(z|x)$ with likelihood distribution $p(x|z)$. The likelihood function $p(x|z)$ is learned with our decoder. The posterior distribution $p(z|x)$ is learned through variational inference by our encoder q. The distribution of z is learned with the re-parameterization trick to ensure model gradient backpropagation by considering our latent space as multivariate Gaussian distributions. The VAE implementation maximizes the evidence of lower bound (ELBO) during training:

$$L(X) = E_{(q(z|x,\phi))}[''log'' p(x|z,\theta)] - KL(q(z|x,\phi)||p(z)) \qquad (2)$$

The variational autoencoder consists of two encoder-decoder networks for proteomic ($AE_p$) and metabolomic ($AE_m$) with a different number of layers and layer embedding sizes. The $AE_p$ consists of a 2-layer encoder of embedding sizes 16 and 8 respectively and a decoder of embedding sizes 16, and 21 (proteomics dimension size). The $AE_m$ consists of a 3-layer encoder of embedding sizes 128, 64, and 32, respectively, and a decoder of embedding sizes 64, 128, and 200 (metabolomics dimension size). The joint embedding $h$ is obtained by concatenating the output of the encoder from $AE_p$ and $AE_m$ feed-forward to a dense layer of embedding size 16. Then $h$ is used to derive the normal distribution with mean $\mu$ and standard deviation $\sigma$. During training, to learn joint embedding from metabolic and protein profiles, we trained two separate encoder-decoder networks: one for metabolic and one for protein profile reconstruction. The two networks share the same embedding space by concatenating the output of the two network encoders and sampling from the same distribution (Supplementary Fig. 53). We compared the joint embedding VAE reconstruction mean absolute error with single modality VAE and showed similar ability for single modality reconstruction with both models (Supplementary Fig. 54a, b). Moreover, when combining lung cancer and tonsil datasets with common metabolite and protein markers, VAE successfully separated the two tissue types (Supplementary Fig. 54c).

## Pseudotime analysis

Single cell pseudotime analysis was performed using protein markers for cell type definition. Two methods, Palentir[71], and Diffusion pseudotime[29] were compared to capture the cell trajectory of B cells inside the GC from protein data. Pseudotime based on single-cell protein marker phenotypes was used to correlate single-cell metabolite across B-cell GC trajectories.

## Statistic and reproducibility

The details of statistical tests employed in each case were provided in the figure captions. All p values were corrected for multiple testing and

the statistical testing method was indicated in the figure captions. We used the following convention to indicate significance with asterisks: not significant (ns) ($P > 0.1$), * ($0.1 > P > 0.01$), ** ($0.01 > P > 0.001$), *** ($0.001 > P > 0.0001$), and **** ($P \leq 0.0001$).

For each tissue type, H&E staining was first performed on sequential tissues to determine the region of interest in imaging. 21 regions from tumor microarray, 11 regions from tonsil slides, and 5 regions from large endometrium slides were used. No data were excluded from the analyses. The experiments were not randomized. The Investigators were not blinded to allocation during experiments and outcome assessment.

### Reporting summary
Further information on research design is available in the Nature Portfolio Reporting Summary linked to this article.

## Data availability
The IMC and 3D-SMF image data generated in this study have been deposited at Zenodo with identifier 6784251. The mass spectrometry proteomics data have been deposited to the ProteomeXchange Consortium via the PRIDE partner repository with the dataset identifier PXD045840. The metabolomic raw data have been deposited to MetaboLights with the dataset identifier MTBLS8685. Source Data are provided in this paper. Source data are provided with this paper.

## Code availability
The analysis codes are available at https://github.com/coskunlab/ScSpaMet. The IMC data was exported using MCD Viewer (v.1.0.560.2). Analysis used Anaconda (v. 4.12.0) and Jupyterlab (v. 3.2.8) with custom-written Python code.

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

## Acknowledgements

A.F.C. holds a Career Award at the Scientific Interface from Burroughs Wellcome Fund and a Bernie-Marcus Early-Career Professorship. A.F.C. was supported by start-up funds from the Georgia Institute of Technology and Emory University. This work was performed in part at the Materials Characterization Facility (MCF) at Georgia Tech. The MCF is jointly supported by the GT Institute for Materials (IMat) and the Institute for Electronics and Nanotechnology (IEN), which is a member of the National Nanotechnology Coordinated Infrastructure supported by the National Science Foundation (Grant ECCS-1542174). This work was performed in part at the Georgia Tech Institute for Electronics and Nanotechnology, a member of the National Nanotechnology Coordinated Infrastructure (NNCI), which is supported by the National Science Foundation (Grant ECCS-2025462). Part of the work was conducted at the Center for Nanophase Materials Sciences, which is a DOE Office of Science User Facility, and using instrumentation within ORNL's Materials Characterization Core provided by UT-Battelle, LLC under Contract No. DE-AC05-00OR22725 with the U.S. Department of Energy. IEN also provided financial support in the form of a Facility Seed Grant. Supported in part by Winship Cancer Institute # IRG-21-137-07 -IRG from the American Cancer Society, American Lung Association Innovation Award, and National Institutes of Health grants (R21AG081715, R21AI173900, and R35GM151028) to A.F.C.

## Author contributions

T.H. and M. Allam equally contributed to the experiments, data analysis, and writing of this paper. S.C., W.H., and A.V.I. assisted in the experiments. B.Y., A.G., M. Afkarian, and S.B. provided tissue samples. A.F.C. supervised the project and wrote the paper.

## Competing interests

The authors declare no competing interests.
