## [Peer Review File NEW · Nature Communications]

Single-cell spatial metabolomics with cell-type specific protein profiling for tissue systems biologyReviewer #1 (Remarks to the Author):

The authors describe a new pipeline named scSpaMet which combines H&E staining, isotope labelling, IMC, 3D-SMF, downstream data integration and interpretation for the multimodal proteomic and metabolic characterization of single cells from patient derived tissue sections. They applied scSpaMet to human lung cancer, tonsil, and endometrium tissue to characterise local metabolic competition using novel spatial metrics, metabolic trajectories and patient level analyte variability.

The study is well designed, well described, innovative and of high interest to a broad audience. Nonetheless, I have minor comments to be addressed.

Comments:

- How is the sample pre-processing (such as H&E staining and isotope labelling) affecting the metabolite composition?
- Single cell segmentation strategy could be improved. I understand the challenge behind segmentation cytosols, but their accurate measurement has deep repercussions on both the cellular metabolic and proteomic expression values and therefore will affect the downstream analysis such as the VAE modelling, metabolic competition and variability measurements.
- Why leave the CD31 expression threshold up to the user? Could it be inferred from the single cell intensity distribution?
- Why was 30um set as the distance to compute the metabolic competition? Could it be expressed in terms of average cell diameter? Alternatively, could this value be derived from the measurements made from the distance to CD31+ cells? This could be a more data-driven way to define it.
- Still in the definition of the metabolic competition, consider the median expression of the surrounding cells rather than the mean as it is less sensitive to extreme values which could be driven by analytical noise.
- How were the metabolites grouped into the high and low-variation channels?
- The github repo was empty at the time of reading.

Optional suggestions to improve on:

Figure 4:

- Align the corresponding fields of view of across b-d and either remove the last one from b or include it in c-d
- Overlapping colormaps in c-d were hard to read
- Heatmaps could have a more quantitative interpretation if paired with another plot (intensity vs distance and associated correlation or R squared values)

Figure 7:

- Show the trajectories from d on the t-sne in b
- B, explain the blue and green colour coding: DZ to LZ seem to match well the blue trajectory but DZ to Activated B matches less the green one as the terminal cluster (4 - purple) is on the right part rather than on the upper branch.
- Minor point: as an alternative visualisation, an embedding on top of a graph directed methods (like PAGA) could be used instead of tSNE
- D, colour code the cells based on their pseudotime values to better visualise the spatial spread of these trajectories.

Curiosity:

- Could IMF be performed before 3D SMF? This could provide clues about their potential adverse effects on each other's readout.
- Can we measure the performance of the VAE to balance the variability between the proteomic and metabolomic modalities?
- How does the cell competition score vary across tissues for the same metabolites?
- Could the VAE be trained on all collected data rather than sample-wise? It could potentially allow for more generalizability.
- How does the VAE handle the differences of dimensionality between the proteomic (21 measures) and the metabolomic (approx 200) modalities? Is this somewhat normalised for when concatenating the two latent vectors which have the same size?

Reviewer #2 (Remarks to the Author):

The authors have developed ScSpaMet, a framework for simultaneous imaging of proteins and metabolites at the single-cell level with high spatial resolution. They applied ScSpaMet to human lung cancer, tonsil, and endometrium tissues, and were able to identify metabolite variations between cells, cell type-specific local metabolic competition, and metabolic trajectories. ScSpaMet provides a complementary solution to the need for simultaneous whole-cell metabolic and protein analysis in situ. It enables accurate identification of various cell types with their corresponding metabolomic profiles and allows high-resolution joint protein and metabolite profiling at the single-cell level in the same tissue. The authors proposed a local cell metabolite competition framework and suggested that ScSpaMet could pave the way to systematic single-cell metabolite and protein profiling in their tissue environment. Overall, the work is technically sound and well-motivated. However, I have some comments before recommending publication.

Major:

1. The mass annotation is a major challenge for TOF-SIMS. It would be interesting to learn how the authors annotated the many metabolites given the integer resolution of m/z values.
2. The simultaneous profiling of proteins and metabolites is quite interesting, even if it is not really in the same cell but sequential tissue. I was hoping to see new biological insights discovered by the combination of metabolites and proteins after the establishment of the technology. However, I was disappointed to see that all the content of the manuscript uses protein profiles to define cell types and use differential analysis to identify known metabolite differences.
3. It might be useful to illustrate how the metabolite abundance is correlated with specific proteins involved in biological pathways, such as glycolysis and the TCA cycle.
4. The zonation pattern around CD31 is interesting. Can the authors provide possible explanations? How do the metabolite zonation patterns correlate with the proteins that were profiled? What is happening at the gene expression level?
5. The distance analysis in terms of CD31 cells is quite interesting but suffers from the potential false discovery rate given the small size of the field of view (FOV) of SIMS. The tissue microenvironment outside the current FOV might have an unexpected impact on the metabolic program. One possible suggestion is to consider the metabolic spatial patterns within certain microenvironments or expand the FOV size of SIMS while reducing the spatial resolution.
6. The patient-level analysis is problematic due to sampling issues and wastes the single-cell resolution provided by the method. A more interesting analysis might be to identify spatial metabolic/proteomic signatures that exhibit specific shapes or metabolic trends of different groups of patients.
7. Figure 7d is interesting to display the direction of cells in terms of the trajectory path. A more insightful discussion of these arrows would be useful.

Minor:

1. Lines 134-136: Why do "single-cell metabolomic profiles have less variation across cell types"?
2. Supplementary Figure 11: The labels and color bars are unclear. Text and legends in figures, especially in Additional and Supplementary Figures, should be more carefully prepared.

Reviewer #3 (Remarks to the Author):

Review for Coskun and coworkers:

In this work Hu et al. present scSpaMet, an approach to measure metabolites in situ and overlay that with measurements of proteins using imaging mass cytometry and metal-labeled antibodies. Altogether, the results seem solid and integration between SIMS-TOF and IMC is nice. That said, my main concern is that throughout the paper results are often reported as "metabolite ## m/z is present more in X than in Y" with no further context or interpretation. I am not an expert in metabolomics and so cannot relate to the standard in the field. However, I suggest for the editor to check whether this conforms with the standard. As an outsider, I find it very hard to follow such results since they have no biological meaning and at the end, I am left wondering "what have I learned from this study?". Even if many of these metabolites are uncharacterized, I think that the authors should do a better job in relating their results to the literature. What is e.g. 48 m/z? What has been shown about it previously? Is there a pathway analysis that can group some of these metabolites together? Following all these measurements, can the authors provide a model of the metabolic processes at play? I am afraid that currently, the level of depth in interpreting the metabolomics results is rather shallow.

Major comments:

1. scSpaMet requires to first measure the metabolites using TOF-SIMS, and then transfer the slide to the IMC to measure protein expression using metal-labeled antibodies. This requires that the same cells be present for both measurements, yet it is my understanding the TOF-SIMS is destructive to the tissue. Could the authors provide information as to how much depth of the tissue is being taken for TOF-SIMS? I could not find this number in the results or methods section, yet I believe that it is important for the readers to know. Especially, since when examining supplementary figures 8-10 it seems that there is some degree of misalignments between the IMC intercalator signal and the SIMS PO3. Could it be that for a good fraction of the cells, the TOF SIMS measurements ablates enough of the tissue such that the metabolic and protein data is actually acquired on different cells?
2. The results in figures 3C,D are unclear. If I understand the analysis correctly, we see metabolites that are present at higher level in e.g. tumor cells, and these differ as a function of distance to blood vessels. For example, Methionine is enriched in tumor cells far from blood vessels, whereas 24m/z is enriched in tumor cells close to vessels. Do the authors have any model for these results? Can they put it in the context of the literature? Otherwise it is very difficult to understand what is the meaning of these results.
3. The images in figures 3C,D are unclear. It is unclear what they are showing. The authors should mark on them the location of blood vessels and the identity of tumor and immune cells such that it is clear how these results relate to the heatmap on the left.
4. The VAE analysis does not appear in any of the main figures or supplementary figure. Only in the extended data. It also does not yield any mentionable results as far as I can see in the relevant section in the results (lines 247 – 268). The authors should do a better job of materially connecting this analysis to the work or discard it.
5. The pseudotime analysis for B-cell differentiation is unclear to me. How do the authors set the starting point? How do they differentiate between DZ to LZ and DZ to activated B? I am missing arrows on the tSNE – what are the bifurcation points?

Minor comments:

6. It is hard to understand how many samples were profiled. The methods claims that the TMA had 21 patients, but in the data it looks like only 7 were profiled. The authors should clearly state in the text how many cores from how many patients were profiled.
7. The use of "extended data" is very confusing. It is unclear to me why some figures are in "supplementary figures" and others are in "extended data". I ask that the authors move all figures that are important for the manuscript to main/supplement as customary.

This document contains our responses to the reviewer. Our answers to the reviewers are highlighted in red. We have also highlighted all changes in the manuscript in red.

Reviewer Comments:

Reviewer #1 (Remarks to the Author):

The authors describe a new pipeline named scSpaMet which combines H&E staining, isotope labelling, IMC, 3D-SMF, downstream data integration and interpretation for the multimodal proteomic and metabolic characterization of single cells from patient derived tissue sections. They applied scSpaMet to human lung cancer, tonsil, and endometrium tissue to characterise local metabolic competition using novel spatial metrics, metabolic trajectories and patient level analyte variability.

The study is well designed, well described, innovative and of high interest to a broad audience. Nonetheless, I have minor comments to be addressed.

We thank the reviewer's comments about our manuscript. We have included more details and taken into account the reviewer's suggestions for improvement.

Comments:

- How is the sample pre-processing (such as H&E staining and isotope labelling) affecting the metabolite composition?

We appreciate the reviewer's comments about pre-processing staining and labeling effect on metabolite composition.

The H&E staining is done on a consecutive tissue slide separately and therefore doesn't affect the metabolite composition. We understand that our description of the process is confusing. We have changed the text in the manuscript to better reflect this: "First, a consecutive tissue slide is stained separately using H&E to identify imaging region of interests before scSpaMet profiling and downstream analysis."

On the other hand, the isotope labeling effect on metabolite composition was comprehensively studied in our 3D-SMF paper¹. We did 3D-SMF metabolite profiling in the same GC from serial tissue sections of tonsil samples. The first tissue sample was labeled by isotope-enrichment targets whereas the second tissue sample with the same anatomical structure was left unlabeled to serve as a control. The total signal of 3D TOF-SIMS per isotope-targeting channels (not naturally found in tissues) showed an increased summed intensity, while the other mass channels either showed enhancement or reduction in the total signal per channel. the signal enhancement due to labeling in each amino acid channel of the labeled and unlabeled GC data, providing a consistent enhancement in all the mass channels for amino acid fragments but doesn't affect the general level of metabolite detection when comparing across channels (3D SMF paper, Supplementary fig. 31-34)

- Single cell segmentation strategy could be improved. I understand the challenge behind segmentation cytosols, but their accurate measurement has deep repercussions on both the cellular metabolic and proteomic expression values and therefore will affect the downstream analysis such as the VAE modelling, metabolic competition and variability measurements.

We agree with the reviewer's comments about the single-cell segmentation strategy could be improved. Because of the low-resolution staining of the cytosol region of single cells, to improve the segmentation of cytosol we used a method combining multiple markers and nuclei markers. First, we use the best available nuclei marker (Histone H3 in tonsil data) and we combine multiple protein markers using maximum projection to get the whole cell area (in tonsil data: CD38, Vimentin, CD21, BCL6, ICOS1, CD11b, CD86, CXCR4, CD11c, FoxP3, CD4, CD138, CXCR5, CD20, CD8, C-Myc, PD1, CD83, Ki67, COL1, CD3, CD27, EZH2). To get the cytosolic region of each cell, we subtract the nuclei image from the combined protein marker image. Finally, we use deepcell Mesmer model for single-cell nuclei and cytosol segmentation. This 2-step segmentation pipeline better captures the overall cytosolic region of single cells by incorporating the majority of available multiplex imaging panels.

We showed in **Supplementary Fig. 50** the overall improvement in capturing cell regions compared to the metabolite channels alone. The combined multiplex images were used for single-cell cytosol and nuclei segmentation (a-c). We compared it to a segmentation using a registered PO3- channel image from 3D-SMF imaging which provided nuclei signals in cells. We showed that the average captured nuclei area using the IMC modality yields higher cell areas compared to 3D-SIMS and 2-fold higher in IMC cytosol segmentation compared to SIMS nuclei.

This pipeline was applied to tonsil tissues where single-cell showed more uniform shape compared to cancer tissues. We have modified our method section in the following: “In tonsil tissues, we use the best available nuclei marker (Histone H3 in tonsil data) and we combine multiple protein markers using maximum projection to get the whole cell area (in tonsil data: CD38, Vimentin, CD21, BCL6, ICOS1, CD11b, CD86, CXCR4, CD11c, FoxP3, CD4, CD138, CXCR5, CD20, CD8, C-Myc, PD1, CD83, Ki67, COL1, CD3, CD27, EZH2). To get the cytosolic region of each cell, we subtract the nuclei image from the combined protein marker image. Finally, we use deepcell² Mesmer model for single-cell nuclei and cytosol segmentation. This 2-step segmentation pipeline better captures the overall cytosolic region of single cells by incorporating the majority of available multiplex imaging panels. The combined multiplex images were used for single-cell cytosol and nuclei segmentation (a-c). We compared it to a segmentation using a registered PO3- channel image from 3D-SMF imaging which provided nuclei signals in cells. We showed that the average captured nuclei area using the IMC modality yields higher cell areas compared to 3D-SIMS and 2-fold higher in IMC cytosol segmentation compared to SIMS nuclei (**Supplementary Fig. 50-51**).”

Supplementary Fig. 50.

Supplementary Fig. 51.

- Why leave the CD31 expression threshold up to the user? Could it be inferred from the single cell intensity distribution?

We thank the reviewer's insight. We agree that the CD31 expression threshold could be inferred from the single-cell intensity distribution. We have changed our pipeline to infer the CD31 expression threshold based on the intensity distribution of CD31 to better reflect CD31-positive cells. We have changed the text in the manuscript to better reflect this: "Single-cell CD31 intensity expression followed a bi-modal Gaussian distribution and cells with higher intensities were defined as CD31+ endothelial cells and validated by inspecting original CD31 marker images." And we have added **Supplementary Fig. 15** to better describe the choice of thresholding.

Supplementary Fig. 15

- Why was 30um set as the distance to compute the metabolic competition? Could it be expressed in terms of average cell diameter? Alternatively, could this value be derived from the measurements made from the distance to CD31+ cells? This could be a more data-driven way to define it.

We thank the reviewer's insight about the distance to compute the metabolic competition. We agree with the reviewer that this could be expressed as a function of average cell diameter. We calculated the average minimum and maximum axis length to infer a mean cell diameter of 10 μm . Therefore a 20 μm was chosen for around 1 cell distance from each cell centroid. In the tonsil sample, because of the high density around the tonsil follicle regions, we have used a contact graph to model metabolite competition. We have added the following text in our method section: "The average minimum and maximum axis length of single cells were used to infer a mean diameter of 10 μm and a distance of 20 μm was chosen based on 1 cell distance"

- Still in the definition of the metabolic competition, consider the median expression of the surrounding cells rather than the mean as it is less sensitive to extreme values which could be driven by analytical noise.

We thank the reviewer's insight about using the median expression of the surrounding cells rather than the mean to take into account the extreme values and noise. We have changed the figures to reflect this.

- How were the metabolites grouped into the high and low-variation channels?

We thank the reviewer's comments on the method used in grouping high and low-variation channels. We used a t-test with overestimated variance in each group to define high and low variation channels across cell types and patients. More specifically, let the sample size, mean and standard deviation of metabolite expression for the case group i be n_i, μ_i, σ_i . The corresponding notations of the control group (cell not in case group i) are n_0, μ_0, σ_0 . The t-test for metabolite is define as follow:

$$\frac{\mu_i - \mu_0}{\sqrt{\frac{\sigma_i^2}{n_i} + \frac{\sigma_0^2}{n_0}}}$$

We added the following text to the manuscript for completeness: "T-test with overestimated variance in each phenotype group is used to define high and low metabolite variation channels across cell types and patients. Let the sample size, mean and standard deviation of metabolite expression for the case group i be n_i, μ_i, σ_i . The corresponding notations of the control group (cell not in case group i) are n_0, μ_0, σ_0 . The t-test for metabolite is define as follow:

$$\frac{\mu_i - \mu_0}{\sqrt{\frac{\sigma_i^2}{n_i} + \frac{\sigma_0^2}{n_0}}}$$

”

- The github repo was empty at the time of reading.

We thank the reviewer's comment about the github repo. We have published our code in the format of jupyter notebooks into the corresponding repo at: <https://github.com/coskunlab/ScSpaMet>

Optional suggestions to improve on:

Figure 4:

- Align the corresponding fields of view of across b-d and either remove the last one from b or include it in c-d

We thank the reviewer's comment about the **Fig. 4** fields of view. We have removed the last FOV from Figure 4. b to match the fields of view shown in Figure 4. c-d. Please see the new Figure 4 below.

- Overlapping colormaps in c-d were hard to read

We agree with the reviewer's comment about colormaps being hard to read in Figure 4. c-d. We have changed it only to show the ratio on the spatial graphs. Please see the new **Fig. 4** below.

- Heatmaps could have a more quantitative interpretation if paired with another plot (intensity vs distance and associated correlation or R squared values)

We thank the reviewer's insight for paring another quantitative plot with the heatmaps in Figure 4. We have added **Supplementary Fig. 19** showing the most and least correlated metabolite channels to distance to CD31+ cells with the corresponding Pearson correlation score.

Supplementary Fig. 19

Figure

7:

- Show the trajectories from d on the t-sne in b

We thank the reviewer's comments and we have added the trajectories from Figure 7. d on the T-SNE in Figure 7. b. We have added **Supplementary Fig. 33**, showing the trajectories on the t-SNE. Please see **Supplementary Fig. 33a** below.

- Minor point: as an alternative visualisation, an embedding on top of a graph directed methods (like PAGA) could be used instead of tSNE

We thank the reviewer's insight on using an embedding on top of graph methods for visualization. We have added a Supplementary Figure showing the trajectories as a graph-directed method. Please see **Supplementary Fig. 33b** below.

- B, explain the blue and green colour coding: DZ to LZ seem to match well the blue trajectory but DZ to Activated B matches less the green one as the terminal cluster (4 -purple) is on the right part rather than on the upper branch.

We thank the reviewer's comments on the color coding of the trajectories. GCs of lymphoid organs are the place where activated B-cells undergo differentiation across Dark Zone (DZ) B-cells, Light Zone (LZ) B cells, Memory B-cells, and Plasma cells. DZ contains the rapidly dividing B-cells undergoing somatic hypermutation (SHM) and LZ contains FDCs, TFH, and B-cells that are exiting the GC area. B-cell migration happens inside GC from DZ to LZ. Moreover, recent studies have provided ample experimental evidence for the re-entry of selected B cells from LZ to DZ upon antigen-driven selection³⁻⁵.

In the right panel of Fig. 7 b, we have shown the pseudotime path (1) the GC DZ B-cells to GC LZ B-cells and (2) the GC DZ B-cells to the activated B-cells. We have changed the colormap to better visualize the trajectories. In the Fig 7b heatmap, both 2-green and 4-purple represent Activated B-cells. We infer that cluster 4 represents a transitional state between Activate B cells and LZ B-cells.

- D, colour code the cells based on their pseudotime values to better visualise the spatial spread of these trajectories.

We thank the reviewer's insight on visualizing the pseudotime values on single cells for a better understanding of the spatial spread of the trajectories. We have added **Supplementary Fig. 34-35** showing the cell colored with their corresponding pseudotime value along trajectories.

Supplementary Fig. 34

Supplementary Fig. 35

Curiosity:

- Could IMF be performed before 3D SMF? This could provide clues about their potential adverse effects on each other's readout.

We thank the reviewer's questions about the order of IMC and 3D-SMF imaging. Both 3D-SMF and IMC utilizes laser ablation on the tissue in order to capture signal by time-of-flight detectors. We chose the order of first imaging using 3D-SMF and then IMC due to the laser ablation power of the machines. In

fact, the TOF-SIMS machine from 3D-SMF has low ablation power allowing areas sputtering rate of 1 micron per hour. On the other hand, IMC uses a high-energy UV laser that ablates through the tissue sample while imaging (see Figure below). Therefore, we first imaged using 3D SMF and then IMC.

- Can we measure the performance of the VAE to balance the variability between the proteomic and metabolomic modalities?

We thank the reviewer's comments on measuring the performance of the VAE. We compared the joint modality embedding VAE model with models using only protein level and metabolic level data. That is, we train the joint VAE by learning two encoder-decoder architectures to learn a joint embedding (left) from the two modalities and compare the reconstruction error with two encoder-decoder VAE architectures learning a metabolite and protein embedding separately (right).

We found a reconstruction mean absolute error of 0.0701 and 0.6103 for protein and metabolomic profiles respectively in tonsil tissues whereas training only a single modality VAE (IMC and SIMS separately) give a score of 0.0813 and 0.6381.

Similarly, we obtained a reconstruction mean absolute error of 0.0822 and 0.5074 for protein and metabolomic profiles respectively in lung cancer tissues whereas training only a single modality VAE (IMC and SIMS separately) give a score of 0.0802 and 0.6201.

Finally, we obtained a reconstruction mean absolute error of 0.1094 and 0.6201 for protein and metabolomic profiles respectively in lung cancer tissues whereas training only a single modality VAE (IMC and SIMS separately) give a score of 0.1019 and 0.6307.

Dataset	Model	Modality	MAE
Tonsil	Joint-VAE	SIMS	0.6103
		IMC	0.0701
	VAE	SIMS only	0.6381
		IMC only	0.0813
Lung	Joint-VAE	SIMS	0.5074
		IMC	0.0822
	VAE	SIMS only	0.5589
		IMC only	0.0802
Endometrium	Joint-VAE	SIMS	0.6201

		IMC	0.1093
	VAE	SIMS only	0.6307
		IMC only	0.1019

- How does the cell competition score vary across tissues for the same metabolites?

We thank the reviewer's insight on cell competition scores across tissues. We have generated a supplementary figure showing the variation of cell competition score across tissues for the same metabolites. We have shown the variation of Tumor/T-Cell competition and Tumor/CD68+ cells across patients (**Supplementary Fig. 22**).

Supplementary Fig. 22

- Could the VAE be trained on all collected data rather than sample-wise? It could potentially allow for more generalizability.

We thank the reviewer on his comment on training the VAE on all collected data rather than sample-wise. We have indeed trained the VAE on all the samples from lung cancer tissues and tonsil tissues respectively. Our lung cancer tissues and tonsil tissues are indeed stained with different antibody panels. We have added a VAE training using a common protein marker to achieve more generalizability as mentioned by the reviewer. The common marker across lung and tonsil datasets are CD20, CD3, CD4, CD8a, Collagen, and Vimentin. Using the VAE low dimensional embedding shows the separation of the lung and tonsil samples.

- How does the VAE handle the differences of dimensionality between the proteomic (21 measures) and the metabolomic (approx 200) modelaities? Is this somewhat normalised for when concatenating the two latent vectors which have the same size?

We thank the reviewer's question on how the VAE handles the differences in dimensionality between proteomics and metabolomics. Because of the difference between the dimensionality of the proteomic and metabolomic measurements as mentioned by the reviewer, our variational autoencoder consists of two encoder-decoder networks for proteomic (AE_p) and metabolomic (AE_m) with a different number of layers and layer embedding sizes. The AE_p consists of a 2-layer encoder of embedding sizes 16 and 8 respectively and a decoder of embedding sizes 16, and 21 (proteomics dimension size). The AE_m consists of a 3-layer encoder of embedding sizes 128, 64, and 32, respectively, and a decoder of embedding sizes 64, 128, and 200 (metabolomics dimension size). The joint embedding h is obtained by concatenating the output of the encoder from AE_p and feed-forward to a dense layer of embedding size 16. Then h is used to derive the normal distribution *with mean* μ and standard deviation σ . The two decoders of AE_p and AE_m use the same low-dimensional embedding to reconstruct the proteomic and metabolomic profiles. We have added this information to the method section for completeness.

Reviewer #2 (Remarks to the Author):

The authors have developed ScSpaMet, a framework for simultaneous imaging of proteins and metabolites at the single-cell level with high spatial resolution. They applied ScSpaMet to human lung cancer, tonsil, and endometrium tissues, and were able to identify metabolite variations between cells, cell type-specific local metabolic competition, and metabolic trajectories. ScSpaMet provides a complementary solution to the need for simultaneous whole-cell metabolic and protein analysis in situ. It enables accurate identification of various cell types with their corresponding metabolomic profiles and allows high-resolution joint protein and metabolite profiling at the single-cell level in the same tissue. The authors proposed a local cell metabolite competition framework and suggested that ScSpaMet could pave the way to systematic single-cell metabolite and protein profiling in their tissue environment. Overall, the work is technically sound and well-motivated. However, I have some comments before recommending publication.

Major:

1. The mass annotation is a major challenge for TOF-SIMS. It would be interesting to learn how the authors annotated the many metabolites given the integer resolution of m/z values.

We thank the reviewer's comments on the method used for annotating many metabolites. We used various literature on TOF-SIMS compound annotation to give a putative annotation of detected metabolite channels. More specifically, the IONTOF SurfaceLab software (version 6) was used to perform basic image processing operations on the acquired spatial mass spectra. The spatial distribution for each selected peak was exported in files containing the coordinates and pixel intensity values. Around 200 peaks were selected using Surface Lab. The data were then exported in American Standard Code for Information Interchange (ASCII) format into a text file. For putative annotation of the metabolite channels, we used a couple of TOF-SIMS metabolite imaging literature. We have also attached the collected putative annotation table in **Supplementary table 8** (ordered by m/z) and **Supplementary table 9** (ordered by

reference). We combined putative annotation from various literature sources which annotates the corresponding m/z from TOF-SIMS to specific molecules which have respective fragmentation sources such as glucose, lipids, amino acids, and cholesterol fragmentation. It has been shown that TOF-SIMS provide a sub-micron lateral resolution with equivalent sensitivity to lipids whereas MALDI and DESI type of imaging mass spectrometry can capture a broader range of biomolecules including proteins, peptides, and nucleotides⁶.

For completeness, we added the following text to the manuscript: “The IONTOF SurfaceLab software (version 6) was used to perform basic image processing operations on the acquired spatial mass spectra. The spatial distribution for each selected peak was exported in files containing the coordinates and pixel intensity values. Around 200 peaks were selected using Surface Lab. The data were then exported in American Standard Code for Information Interchange (ASCII) format into a text file. For putative annotation of the metabolite channels, we used published work of TOF-SIMS metabolite imaging literature (Supplementary Table. 8-9).”

Moreover, we agree with the reviewer’s comment on the major challenge for TOF-SIMS mass annotation. In order to provide an alternative to the annotation task, we have used the METASAPCE⁷ platform to provide annotation of mass by cross-referencing all the public datasets provided on the platform. This platform provides an automated annotation pipeline that can provide alternative mass annotation suggestions for any user to explore. We have uploaded example data from our 3D-SMF pipeline to the following address: <https://metaspace2020.eu/project/hu-2023>.

2. The simultaneous profiling of proteins and metabolites is quite interesting, even if it is not really in the same cell but sequential tissue. I was hoping to see new biological insights discovered by the combination of metabolites and proteins after the establishment of the technology. However, I was disappointed to see that all the content of the manuscript uses protein profiles to define cell types and use differential analysis to identify known metabolite differences.

We thank the reviewer’s insights on the discovery of the combination of metabolites and proteins. In our imaging pipeline, metabolite and protein are imaged simultaneously on the same tissue, therefore achieving the same cell profiling. We understand the phrasing in the manuscript might be misleading and we have changed it to: “First, a consecutive tissue slide is stained separately using H&E to identify imaging region of interests before scSpaMet profiling and downstream analysis (**Supplementary Fig. 2a and Supplementary Fig. 3-4**). Next, sequential ToF-SIMS and IMC imaging are performed to extract spatial maps of metabolites and proteins **in the same tissue**. Pixel clustering of SIMS data reveals unique metabolite variation in the spatial context (**Supplementary Fig. 5-7 and Methods**).”

On the other hand, we agree with the reviewer's comments on the discovery of new biological insights from a combination of metabolites and proteins. We have proposed a VAE framework for the characterization of joint protein metabolic states in situ. We compared the joint modality embedding VAE

model with models using only protein level and metabolic level data. We found a reconstruction mean square error of 0.0792 and 0.4997 for protein and metabolomic profiles respectively in lung cancer tissues whereas training only a single modality VAE (IMC and SIMS separately) give a score of 0.0768 and 0.5230. Similarly, we obtained a reconstruction mean square error of 0.0941 and 0.6257 for protein and metabolomic profiles respectively in tonsil tissues whereas training only a single modality VAE (IMC and SIMS separately) give a score of 0.0809 and 0.6372.

3. It might be useful to illustrate how the metabolite abundance is correlated with specific proteins involved in biological pathways, such as glycolysis and the TCA cycle.

We thank the reviewer's insights on correlating metabolite abundance with specific proteins. We have added a correlation of metabolites plot for each protein marker in lung cancer and tonsil tissues as **Supplementary Fig. 12** and **Supplementary Fig. 26**. Moreover, we have added **Supplementary Fig. 13** and **Supplementary Fig 27** showing the correlation of glycolysis and TCA cycle associated channels in lung cancer and tonsil tissues.

Channel	Category	Type
71	Glycolysis	Glucose fragment
87	Glycolysis	Glucose fragment
99	Glycolysis	Glucose fragment
119	Glycolysis	Glucose fragment
141	Glycolysis	Glucose fragment
159	Glycolysis	Glucose fragment
177	Glycolysis	Glucose fragment
179	Glycolysis	Glucose fragment
359	Glycolysis	Glucose fragment
199.17	Fatty Acid	FA(12:0)
227.18	Fatty Acid	FA(14:0)
251.2	Fatty Acid	FA(16:2)
253.2	Fatty Acid	FA(16:1)
255.23	Fatty Acid	FA(16:0)
275.2	Fatty Acid	FA(18:4)
277.21	Fatty Acid	FA(18:3)
279.22	Fatty Acid	FA(18:2)
281.25	Fatty Acid	FA(18:1)
283.35	Fatty Acid	FA(18:0)

Supplementary Fig. 12

Supplementary Fig. 13

Supplementary Fig. 26

Supplementary Fig. 27

4. The zonation pattern around CD31 is interesting. Can the authors provide possible explanations? How

do the metabolite zonation patterns correlate with the proteins that were profiled? What is happening at the gene expression level?

We thank the reviewer's comments on CD31 zonation pattern. CD31 marker is typically expressed in endothelial cells around vessels⁸. It has been shown that antigens are transported and presented by vascular cells and initiate rapid and localized memory immune response⁹. Moreover, cancer cells need a large amount of nutrients to support growth by the vascular system, and metabolic differences have been observed between tumor and non-tumor regions¹⁰. Therefore, we herein present a framework for quantifying metabolic variation at the single-cell level as distance to endothelial cells.

5. The distance analysis in terms of CD31 cells is quite interesting but suffers from the potential false discovery rate given the small size of the field of view (FOV) of SIMS. The tissue microenvironment outside the current FOV might have an unexpected impact on the metabolic program. One possible suggestion is to consider the metabolic spatial patterns within certain microenvironments or expand the FOV size of SIMS while reducing the spatial resolution.

We thank the reviewer's comments on CD31 distance analysis that might suffer from the potential false discovery rate given the small size of the field of view. We enlarge the discovery of CD31-positive cells to the whole Tumor microarray core and perform the same analysis. For each imaging region, we have matched the original location inside the tumor microarray (TMA) core images (**Supplementary Figure 16-17**). After matching each imaging region with its corresponding single-cell segmentation mask, we superpose the original CD31 marker imaging on the TMA core images. We extract CD31 positive regions from CD31 marker images and for each pixel inside the TMA core image, we calculate the minimum distance to a CD31 positive region. Therefore, we obtained a spatial distance map from CD31 positive regions for each TMA core image. Each corresponding protein and metabolic imaging region have a single cell distance to the CD31 positives region by matching each cell in their original TMA core location.

We have added the following text in the manuscript: "First, by leveraging CD31 protein markers from IMC multiplex data, we defined CD31+ endothelial cells in each patient tissue image. To consider the small size of the imaging field of view (FOV) in the 3D-SMF pipeline, we used the whole TMA core IMC images to detect CD31-positive cells. Single-cell CD31 intensity expression followed a bi-modal Gaussian distribution and cells with higher intensities were defined as CD31+ endothelial cells and validated by inspecting original CD31 marker images (Supplementary Fig 15). We matched each 3D-SMF image region back into the IMC images (Supplementary Fig. 16-17). Then for each segmented cell in the 3D-SMF image region, the minimum distance to CD31+ endothelial cells is extracted by identifying the cell centroids' position closest to the CD31+ endothelial cells from larger IMC CD31 images by kNN search with spatial data of single-cells (Fig. 5a and Supplementary Fig. 16-18)."

We have replaced Fig. 4 images with the newly defined CD31 distance maps.

Supplementary Figure 16

Supplementary Figure 17

6. The patient-level analysis is problematic due to sampling issues and wastes the single-cell resolution provided by the method. A more interesting analysis might be to identify spatial metabolic/proteomic signatures that exhibit specific shapes or metabolic trends of different groups of patients.

We thank the reviewer's insights on analyzing the metabolic trends of different groups of patients. We agree with the reviewer that the analysis of spatial metabolic/proteomic signatures for different groups of patients will yield a more interesting result. In order to generate spatial signatures incorporating joint metabolomic and proteomics levels, we obtain unsupervised clustering labels for each cell based on their VAE latent embedding and we extract each cell neighboring information by taking a radius threshold of $25 \mu\text{m}$. For each cell, we count the unsupervised cluster labels in its neighborhood and obtain a vector corresponding to the count of each clustering type around its neighborhood. After normalization of the vectors by their total count (with a density equal to one per cell), we perform a second unsupervised

clustering which takes into account the spatial signature of joint metabolic and proteomics profiles. We have added **Fig 6** showing the distribution of spatial signatures across lung cancer patients.

Fig 6.

Supplementary Fig 29

7. Figure 7d is interesting to display the direction of cells in terms of the trajectory path. A more insightful discussion of these arrows would be useful.

We agree with the reviewer's comment on the discussion of cell direction in terms of the trajectory path. We added the following text to the manuscript for completeness: "GCs of lymphoid organs are the place where activated B-cells undergo differentiation across Dark Zone (DZ) B-cells, Light Zone (LZ) B cells, Memory B-cells, and Plasma cells. DZ contains the rapidly dividing B-cells undergoing somatic hypermutation (SHM) and LZ contains FDCs, TFH, and B-cells that are exiting the GC area. B-cell migration happens inside GC from DZ to LZ. Moreover, recent studies have provided ample experimental evidence for the re-entry of selected B cells from LZ to DZ upon antigen-driven selection. We inferred DZ to LZ and DZ to Activated B cell trajectories from protein markers. We represented cell type and their corresponding trajectory back into their spatial domain showing the polarity of B cell inside the germinal center and spatial migration pattern along cell differentiation trajectories."

Minor:

1. Lines 134-136: Why do "single-cell metabolomic profiles have less variation across cell types"?

We thank the reviewer's question on the "single-cell metabolomics profiles have less variation across cell types". The single-cell metabolic expression captured by 3D-SMF has lower variation across cells compared to protein-level expression imaged using IMC. This might be explained by the abundance of metabolite shared across cells compared to the specific proteins present for specific cell types. Moreover, it is not clear how well metabolites in cells are preserved using the protein fixation protocol while protein fixation is well established. Finally, the single-cell imaging resolution using 3D-SMF will detect fewer ions compared to other imaging technologies such as MALDI or low-resolution TOF-SIMS¹¹.

2. Supplementary Figure 11: The labels and color bars are unclear. Text and legends in figures, especially in Additional and Supplementary Figures, should be more carefully prepared.

We thank the reviewer's comment on the text and legends in the figures. We have updated the Additional and Supplementary Figures with high-resolution versions.

Reviewer #3 (Remarks to the Author):

Review for Coskun and coworkers:

In this work Hu et al. present scSpaMet, an approach to measure metabolites in situ and overlay that with measurements of proteins using imaging mass cytometry and metal-labeled antibodies. Altogether, the results seem solid and integration between SIMS-TOF and IMC is nice. That said, my main concern is that throughout the paper results are often reported as "metabolite ## m/z is present more in X than in Y" with no further context or interpretation. I am not an expert in metabolomics and so cannot relate to the standard in the field. However, I suggest for the editor to check whether this conforms with the standard. As an outsider, I find it very hard to follow such results since they have no biological meaning and at the

end, I am left wondering “what have I learned from this study?”. Even if many of these metabolites are uncharacterized, I think that the authors should do a better job in relating their results to the literature. What is e.g. 48 m/z? What has been shown about it previously? Is there a pathway analysis that can group some of these metabolites together? Following all these measurements, can the authors provide a model of the metabolic processes at play? I am afraid that currently, the level of depth in interpreting the metabolomics results is rather shallow.

Major comments:

1. scSpaMet requires to first measure the metabolites using TOF-SIMS, and then transfer the slide to the IMC to measure protein expression using metal-labeled antibodies. This requires that the same cells be present for both measurements, yet it is my understanding the TOF-SIMS is destructive to the tissue. Could the authors provide information as to how much depth of the tissue is being taken for TOF-SIMS? I could not find this number in the results or methods section, yet I believe that it is important for the readers to know. Especially, since when examining supplementary figures 8-10 it seems that there is some degree of misalignments between the IMC intercalator signal and the SIMS PO3. Could it be that for a good fraction of the cells, the TOF SIMS measurements ablates enough of the tissue such that the metabolic and protein data is actually acquired on different cells?

We thank the reviewer’s comments on the method used for annotating many metabolites. Both 3D-SMF and IMC utilizes laser ablation on the tissue in order to capture signal by time-of-flight detectors. We chose the order of first imaging using 3D-SMF and then IMC due to the laser ablation power of the machines. In fact, the TOF-SIMS machine from 3D-SMF has low ablation power allowing areas sputtering rate of 1 micron per hour. On the other hand, IMC uses a high-energy UV laser that ablates through the tissue sample while imaging. Therefore, we first imaged using 3D SMF and then IMC.

We added the following text to the manuscript for completeness: “Depth profiling in the TOF-SIMS was performed around 30-40 slides for 10s ablation at 2-kV per slice for an estimate ablation of 1 micron per hour.”

2. The results in figures 4C,D are unclear. If I understand the analysis correctly, we see metabolites that are present at higher level in e.g. tumor cells, and these differ as a function of distance to blood vessels. For example, Methionine is enriched in tumor cells far from blood vessels, whereas 24m/z is enriched in tumor cells close to vessels. Do the authors have any model for these results? Can they put it in the context of the literature? Otherwise it is very difficult to understand what is the meaning of these results.

We thank the reviewer for the comment in Figure 4. C and D. As the reviewer mentioned, the analysis is showing the metabolite competition between the tumor cells and immune cells as a distance to blood vessels. SEAM paper showed metabolic variation around the central vein (CV) in mouse liver tissues with selected mass channels such as m/z 58.00, 59.01, 69.00, 71.02, 87.0, and 101.03 were enriched around the

CV¹². We aim to quantify the metabolite distribution in cell neighborhoods containing tumor and immune cells as a function of distance to blood vessels. CD31 marker is typically expressed in endothelial cells around vessels⁸. It has been shown that antigens are transported and presented by vascular cells and initiate rapid and localized memory immune response⁹. Moreover, cancer cells need a large number of nutrients to support growth by the vascular system, and metabolic differences have been observed between tumor and non-tumor regions¹⁰. Therefore, we herein present a framework for quantifying metabolic variation at the single-cell level as distance to endothelial cells.

3. The images in figures 4C,D are unclear. It is unclear what they are showing. The authors should mark on them the location of blood vessels and the identity of tumor and immune cells such that it is clear how these results relate to the heatmap on the left.

We agree with the reviewer that the images in Figure 4. C and D are unclear. We have changed the figure to better show the corresponding tumor and immune cells. Moreover, we have added a supplementary Figure to show the CD31-positive cells in their spatial context.

4. The VAE analysis does not appear in any of the main figures or supplementary figure. Only in the extended data. It also does not yield any mentionable results as far as I can see in the relevant section in the results (lines 247 – 268). The authors should do a better job of materially connecting this analysis to the work or discard it.

We thank the reviewer for the comments about VAE analysis. We have updated the figure and text to better incorporate the discussion in this study.

We compared the joint modality embedding VAE model with models using only protein level and metabolic level data. That is, we train the joint VAE by learning two encoder-decoder architectures to learn a joint embedding (left) from the two modalities and compare the reconstruction error with two encoder-decoder VAE architectures learning a metabolite and protein embedding separately (right).

We found a reconstruction mean absolute error of 0.0701 and 0.6103 for protein and metabolomic profiles respectively in tonsil tissues whereas training only a single modality VAE (IMC and SIMS separately) give a score of 0.0813 and 0.6381.

Similarly, we obtained a reconstruction mean absolute error of 0.0822 and 0.5074 for protein and metabolomic profiles respectively in lung cancer tissues whereas training only a single modality VAE (IMC and SIMS separately) give a score of 0.0802 and 0.6201.

Finally, we obtained a reconstruction mean absolute error of 0.1094 and 0.6201 for protein and metabolomic profiles respectively in lung cancer tissues whereas training only a single modality VAE (IMC and SIMS separately) give a score of 0.1019 and 0.6307.

Dataset	Model	Modality	MAE
Tonsil	Joint-VAE	SIMS	0.6103
		IMC	0.0701
	VAE	SIMS only	0.6381
		IMC only	0.0813
Lung	Joint-VAE	SIMS	0.5074
		IMC	0.0822
	VAE	SIMS only	0.5589
		IMC only	0.0802
Endometrium	Joint-VAE	SIMS	0.6201

VAE	IMC	0.1093
	SIMS only	0.6307
	IMC only	0.1019

Next, we investigated spatial metabolic/proteomic signatures for different groups of patients. In order to generate spatial signatures incorporating joint metabolomic and proteomics levels, we obtain unsupervised clustering labels for each cell based on their VAE latent embedding and we extract each cell neighboring information by taking a radius threshold of $25 \mu\text{m}$. For each cell, we count the unsupervised cluster labels in its neighborhood and obtain a vector corresponding to the count of each clustering type around its neighborhood. After normalization of the vectors by their total count (with a density equal to one per cell), we perform a second unsupervised clustering which takes into account the spatial signature of joint metabolic and proteomics profiles. We have added **Fig 6** showing the distribution of spatial signatures across lung cancer patients.

Fig. 6

5. The pseudotime analysis for B-cell differentiation is unclear to me. How do the authors set the starting point? How do they differentiate between DZ to LZ and DZ to activated B? I am missing arrows on the tSNE – what are the bifurcation points?

We thank the reviewer for his comments on pseudotime analysis for B-cell differentiation. GCs of lymphoid organs are the place where activated B-cells undergo differentiation across Dark Zone (DZ) B-cells, Light Zone (LZ) B cells, Memory B-cells, and Plasma cells. DZ contains the rapidly dividing B-cells undergoing somatic hypermutation (SHM) and LZ contains FDCs, TFH, and B-cells that are exiting the GC area. B-cell migration happens inside GC from DZ to LZ. Moreover, recent studies have provided ample experimental evidence for the re-entry of selected B cells from LZ to DZ upon antigen-driven selection³⁻⁵. Here we infer single-cell hierarchy from their protein profiles. We selected the starting point from single-cell expression corresponding to Dark Zone B-cells. After plotting the cells in the embedding space (**Fig. 9b**), we determine two distinct trajectories that represented DZ to LZ and DZ to activated B cells from single-cell protein expression across trajectories and their corresponding diffusion pseudo time value. The bifurcation point was determined empirically by the embedding plot of the single-cells. We have added the arrow on the TSNE in the **Supplementary Fig. 33**.

Supplementary Fig. 33

Minor

comments:

6. It is hard to understand how many samples were profiled. The methods claims that the TMA had 21 patients, but in the data it looks like only 7 were profiled. The authors should clearly state in the text how many cores from how many patients were profiled.

We thank the reviewer for his comments on how many samples were profiled. We have imaged 21 regions of interest from 7 patients. We have changed the text in the manuscript to better reflect this: “Patients’ samples for lung tumor were obtained from a tumor microarray (TMA) purchased from a third-party vendor (Biomax, US) with the tissue ID: BS04081a. This TMA included a total of 63 tissue cores of formalin-fixed paraffin-embedded (FFPE) non-small cell lung adenocarcinoma and adjacent normal lung tissue samples obtained from 21 patients. We imaged 21 regions of interest from 7 cores. Each tissue core had a diameter of 1 mm and a thickness of 5- μ m which is within the tissue thickness recommended for IMC (≤ 7 - μ m). The tissue labeling protocol was followed as previously reported in the protocol including antigen retrieval, protein blocking, metal-tagged antibody labeling, and nucleus counterstains. After the staining process is complete, the stained tissues were stored at 4C until imaging time. The human tonsil tissue sections were from TissueArray.com under the IDs HuFPT161. Tonsil sample 1 had tissue ID SU1 and tonsil sample 2 had tissue ID SM2. Tonsil sample 1 had 5 imaging regions of interest and tonsil sample 2 had 6 imaging regions of interest.”

7. The use of “extended data” is very confusing. It is unclear to me why some figures are in “supplementary figures” and others are in “extended data”. I ask that the authors move all figures that are important for the manuscript to main/supplement as customary.

We thank the reviewer for his comments on the use of “Extended data”. We have reorganized our figures into 10 main figures and supplementary figures.

References

1. Ganesh, S. *et al.* Spatially resolved 3D metabolomic profiling in tissues. *Sci. Adv.* **7**, (2021).
2. Greenwald, N. F. *et al.* Whole-cell segmentation of tissue images with human-level performance using large-scale data annotation and deep learning.
<http://biorxiv.org/lookup/doi/10.1101/2021.03.01.431313> (2021) doi:10.1101/2021.03.01.431313.
3. Allen, C. D. C., Okada, T., Tang, H. L. & Cyster, J. G. Imaging of Germinal Center Selection Events During Affinity Maturation. *Science* **315**, 528–531 (2007).

4. Gitlin, A. D., Shulman, Z. & Nussenzweig, M. C. Clonal selection in the germinal centre by regulated proliferation and hypermutation. *Nature* **509**, 637–640 (2014).
5. Schwickert, T. A. *et al.* In vivo imaging of germinal centres reveals a dynamic open structure. *Nature* **446**, 83–87 (2007).
6. Passarelli, M. K. & Winograd, N. Lipid imaging with time-of-flight secondary ion mass spectrometry (ToF-SIMS). *Biochim. Biophys. Acta BBA - Mol. Cell Biol. Lipids* **1811**, 976–990 (2011).
7. Palmer, A. *et al.* FDR-controlled metabolite annotation for high-resolution imaging mass spectrometry. *Nat. Methods* **14**, 57–60 (2017).
8. Lertkiatmongkol, P., Liao, D., Mei, H., Hu, Y. & Newman, P. J. Endothelial functions of platelet/endothelial cell adhesion molecule-1 (CD31). *Curr. Opin. Hematol.* **23**, 253–259 (2016).
9. Pober, J. S., Merola, J., Liu, R. & Manes, T. D. Antigen Presentation by Vascular Cells. *Front. Immunol.* **8**, 1907 (2017).
10. Lidonnici, J., Santoro, M. M. & Oberkersch, R. E. Cancer-Induced Metabolic Rewiring of Tumor Endothelial Cells. *Cancers* **14**, 2735 (2022).
11. Dolatmoradi, M., Samarah, L. Z. & Vertes, A. Single-Cell Metabolomics by Mass Spectrometry: Opportunities and Challenges. *Anal. Sens.* anse.202100032 (2021) doi:10.1002/anse.202100032.
12. Yuan, Z. *et al.* SEAM is a spatial single nuclear metabolomics method for dissecting tissue microenvironment. *Nat. Methods* **18**, 1223–1232 (2021).

Reviewer #1 (Remarks to the Author):

The authors have convincingly addressed all the concerns and points I raised in my initial review. Their detailed responses and explanations are commendable, and they've also effectively integrated the diverse suggestions provided.

Based on these revisions and the current quality of the work, I believe the study is now suitable for publication in Nature Communications.

Reviewer #2 (Remarks to the Author):

I am very impressed by the additional work, especially the data of enlarged FOVs and additional analysis on the correlations across omics. Congratulations on this great work!

Reviewer #3 (Remarks to the Author):

The authors have clearly taken time to address the comments rigorously, and the method in itself is interesting. Unfortunately, my main concern was left unanswered, which is what new biology did we learn from this manuscript? Most metabolic results are reported as ## m/z and are not connected to known metabolites, pathways and functions. The discussion of the results is superficial. This observation is also mirrored in comment 2 of reviewer 2. The authors responded to reviewer 2's comment by relating to the VAE analysis, but I have failed to understand what novel biology is revealed by this analysis, other than saying that metabolism associates with cell types. I still believe that the manuscript is appropriate for Nature Communications. The method is interesting and an important step in connecting protein expression and metabolomics in situ. It is a shame that the work does not produce more biological findings.

REVIEWERS' COMMENTS

Reviewer #1 (Remarks to the Author):

The authors have convincingly addressed all the concerns and points I raised in my initial review. Their detailed responses and explanations are commendable, and they've also effectively integrated the diverse suggestions provided.

Based on these revisions and the current quality of the work, I believe the study is now suitable for publication in Nature Communications.

We thank the reviewer's comments on our manuscript and revision.

Reviewer #2 (Remarks to the Author):

I am very impressed by the additional work, especially the data of enlarged FOVs and additional analysis on the correlations across omics. Congratulations on this great work!

We thank the reviewer's comments on our manuscript and revision.

Reviewer #3 (Remarks to the Author):

The authors have clearly taken time to address the comments rigorously, and the method in itself is interesting. Unfortunately, my main concern was left unanswered, which is what new biology did we learn from this manuscript? Most metabolic results are reported as ## m/z and are not connected to known metabolites, pathways and functions. The discussion of the results is superficial. This observation is also mirrored in comment 2 of reviewer 2. The authors responded to reviewer 2's comment by relating to the VAE analysis, but I have failed to understand what novel biology is revealed by this analysis, other than saying that metabolism associates with cell types. I still believe that the manuscript is appropriate for Nature Communications. The method is interesting and an important step in connecting protein expression and metabolomics in situ. It is a shame that the work does not produce more biological findings.

We thank the reviewer's comments on our manuscript and revision. We agree with the reviewers that some biological insights are unanswered in the manuscript. To address this, we have added Supplementary Figure 38 summarizing our biological findings and relating the metabolic results to known metabolite, pathways and functions when available.

We believe that our framework will provide a method for studying single-cell level metabolite profiles correlated to protein expression in situ.